# Spatially and temporally defined lysosomal leakage facilitates mitotic chromosome segregation

Saara Hämälistö[1,8], Jonathan Lucien Stahl [1,8], Elena Favaro[1], Qing Yang [1], Bin Liu [1], Line Christoffersen[1], Ben Loos [2], Claudia Guasch Boldú[3], Johanna A. Joyce [4], Thomas Reinheckel [5,6], Marin Barisic [3,7] & Marja Jäättelä [1,7]*

Lysosomes are membrane-surrounded cytoplasmic organelles filled with a powerful cocktail of hydrolases. Besides degrading cellular constituents inside the lysosomal lumen, lysosomal hydrolases promote tissue remodeling when delivered to the extracellular space and cell death when released to the cytosol. Here, we show that spatially and temporally controlled lysosomal leakage contributes to the accurate chromosome segregation in normal mammalian cell division. One or more chromatin-proximal lysosomes leak in the majority of prometaphases, after which active cathepsin B (CTSB) localizes to the metaphase chromatin and cleaves a small subset of histone H3. Stabilization of lysosomal membranes or inhibition of CTSB activity during mitotic entry results in a significant increase in telomere-related chromosome segregation defects, whereas cells and tissues lacking CTSB and cells expressing CTSB-resistant histone H3 accumulate micronuclei and other nuclear defects. These data suggest that lysosomal leakage and chromatin-associated CTSB contribute to proper chromosome segregation and maintenance of genomic integrity.

[1] Cell Death and Metabolism, Center for Autophagy, Recycling and Disease, Danish Cancer Society Research Center, 2100 Copenhagen, Denmark. [2] Department of Physiological Sciences, Stellenbosch University, 7600 Stellenbosch, South Africa. [3] Cell Division and Cytoskeleton, Danish Cancer Society Research Center, 2100 Copenhagen, Denmark. [4] Ludwig Institute for Cancer Research, University of Lausanne, 1005 Lausanne, Switzerland. [5] Institute of Molecular Medicine and Cell Research, Medical Faculty, University of Freiburg, 79104 Freiburg, Germany. [6] German Cancer Consortium (DKTK) and German Cancer Research Center (DKFZ), Heidelberg, partner site Freiburg, 79106 Freiburg, Germany. [7] Department of Cellular and Molecular Medicine, Faculty of Health Sciences, University of Copenhagen, 2200 Copenhagen, Denmark. [8] These authors contributed equally: Saara Hämälistö, Jonathan Lucien Stahl *email: mj@cancer.dk

Late endosomes and lysosomes (hereafter referred to as lysosomes) are acidic, membrane-surrounded organelles that function as primary "recycling stations" of eukaryotic cells[1–3]. Inside the acidic lysosomal lumen, over 50 hydrolases ensure the digestion of cellular macromolecules delivered to them by endocytic, autophagic and phagocytic pathways to breakdown products available for metabolic reutilization. In addition to their degradative and recycling roles, lysosomes are emerging as metabolic signaling hubs that adjust cellular metabolism according to the availability of energy and macromolecules by regulating the activities of several metabolic kinases (e.g., mTORC1 and AMPK) and transcription factors (e.g., TFEB, TFE3, and STAT3) located on their surface[4–8].

Besides their established function in the terminal degradation inside the lysosomal lumen, lysosomal hydrolases, especially proteases of cysteine cathepsin family, have been assigned several extracellular functions, e.g., in bone resorption, tumor invasion, and tumor angiogenesis. These processes involve tightly controlled $Ca^{2+}$-regulated lysosomal exocytosis and the release of lysosomal contents to the extracellular space[9], where cysteine cathepsins cleave specific substrates, including components of the extracellular matrix, other proteases, chemokines and E-cadherin[10–12]. Furthermore, lysosomes with their capacity to digest entire organelles and cells can be considered as ticking suicide bombs. Lysosomal membrane permeabilization serves as a trigger for an evolutionarily conserved lysosome-dependent cell death program, which contributes to numerous physiological processes (e.g., immune tolerance and mammary gland involution), as well as to degenerative, inflammatory and microbial diseases and cancer therapy[13–16]. Whereas drugs with lysosomotropic detergent properties, viral entry proteins, and microbial toxins form pores in the endolysosomal membranes[17–19], the mechanisms underlying lysosomal membrane permeabilization induced by other lysosome disrupting stimuli are as yet equivocal. Suggested second messengers include reactive oxygen species[20], free fatty acids[21], and lipid metabolites[22], while lysosomal-associated membrane proteins LAMP1 and LAMP2[23], acid sphingomyelinase[24], dihydroceramide desaturase 1[25] and cholesterol[26] have a stabilizing effect on the lysosomal membrane. Upon intracellular lysosomal leakage, lysosomal hydrolases enter the cytosol, where cathepsins can initiate apoptotic or non-apoptotic cell death pathways[27,28]. With respect to their extra-lysosomal functions, it is important to note that in spite of their acidic pH optimum around 4.5, most cysteine cathepsins retain their activity in neutral pH for a short while, albeit with changed enzyme kinetics and substrate specificity[11,12].

In order to create new, functional cells to eventually replace the dying ones, somatic cells progress through a tightly controlled cell-division cycle, which can be divided into interphase, mitosis, and cytokinesis[29]. During interphase, cells grow in size, replicate their DNA and synthesize other macromolecules and organelles. The five phases of mitosis are characterized by (i) the initial chromatin condensation in prophase, (ii) nuclear membrane disintegration and initiation of mitotic spindle formation in prometaphase, (iii) alignment of chromosomes in the equator of the cell in metaphase, (iv) separation of sister chromatids to two daughter cells in anaphase, and (v) reformation of nuclear envelopes in telophase. Finally, the daughter cells separate from each other in cytokinesis. Mitosis must be carried out accurately to avoid mistakes in chromosome segregation, which can lead to chromosome rearrangements, loss of chromosome fragments or alterations in chromosome copy number. Such errors are likely to disrupt tissue homeostasis and lead to cell death or contribute to accelerated genomic instability characteristic of aging and cancer[30,31]. Accordingly, mitotic chromosome segregation is tightly monitored by the mitotic checkpoint that

delays the irreversible chromosome segregation until the chromosomes are correctly attached to the mitotic spindle, each sister chromatid pair having stable attachments to both poles of the mitotic spindle through their kinetochore complexes at the centromeric region. Even though the cell division cycle is a tightly controlled, errors in chromosome segregation occur, especially in cancer cells[32]. Chromosome segregation errors detectable in anaphase, such as lagging chromosomes, chromatin bridges, and acentric chromatin fragments, can be caused by incomplete chromosome replication, telomere attrition, unresolved DNA damage repair intermediates and intertwined sister chromatids resulting from replication stress in the S phase of the cell division cycle or incorrect kinetochore-microtubule attachments frequently established during early mitosis[30,33–36]. Lagging chromosomes and acentric chromatin fragments arising from breaking chromatin bridges segregate either to the proper or incorrect daughter cell. Due to their delayed segregation, they often fail to incorporate into the newly formed nucleus and form so-called micronuclei, whose reintegration into the genome causes chromosomal instability that in its most extreme cases can lead to an extensive rearrangement of chromosomes[37,38].

Due to the strong degradative potential of lysosomes, it has been widely assumed that the maintenance of lysosomal membrane integrity is a prerequisite for cell viability. The recently developed highly sensitive galectin puncta assay, which can detect even a minor lysosomal leakage, has, however, challenged this dogma[39]. This assay takes advantage of the high affinity binding of galectins (e.g., galectin-3/LGALS3) to the β-galactoside-rich glycocalyx that decorates the luminal side of lysosomal membranes[39,40]. Upon lysosomal membrane permeabilization, cytosolic galectins enter lysosomal lumen and bind β-galactoside with high affinity, thereby leaving a mark on leaky lysosomes that can be visualized even several hours after the lysosomal membrane has regained integrity[39]. This sensitive method has revealed that cells can tolerate a certain degree of lysosomal leakage, which has opened the intriguing possibility that akin to the lysosomal exocytosis which delivers lysosomal hydrolases to the extracellular space, controlled the intracellular release of lysosomal enzymes may carry some, as yet unknown, physiological functions[41].

Here, we demonstrate that leakage of cathepsin B (CTSB) protease from chromatin-proximal leaky lysosomes in prometaphase assists mitotic chromosome segregation in mammalian cells and tissues. Thus, controlled intracellular release of lysosomal CTSB, indeed, serves an essential physiological function during mitosis.

## Results

**Leakage of chromatin-proximal lysosomes during mitosis.** Employing antibodies against LGALS3 and lysosomal-associated membrane proteins 1 (LAMP1) or 2 (LAMP2), we have previously demonstrated that most of the MCF7 breast carcinoma and U2OS osteosarcoma cells growing in optimal culture conditions are, as expected, devoid of LGALS3-positive, leaky lysosomes[39]. While optimizing the staining procedure, we realized, however, that >50% of metaphase cells had a few weakly LGALS3-positive lysosomes in the close proximity of the condensed chromatin (Fig. 1a, b; Supplementary Fig. 1a). Super-resolution structured illumination microscopy (SR-SIM) confirmed that the LGALS3-positive lysosomes were in contact with metaphase chromatin (Fig. 1c). Notably, >80% of the leaky lysosomes co-localized with telomeric repeat binding factor 1 (TERF1) (Fig. 1d and Supplementary Fig. 1b), which binds with high affinity to the duplex telomeric DNA sequences throughout the cell cycle[36]. On the contrary, practically none of the leaky

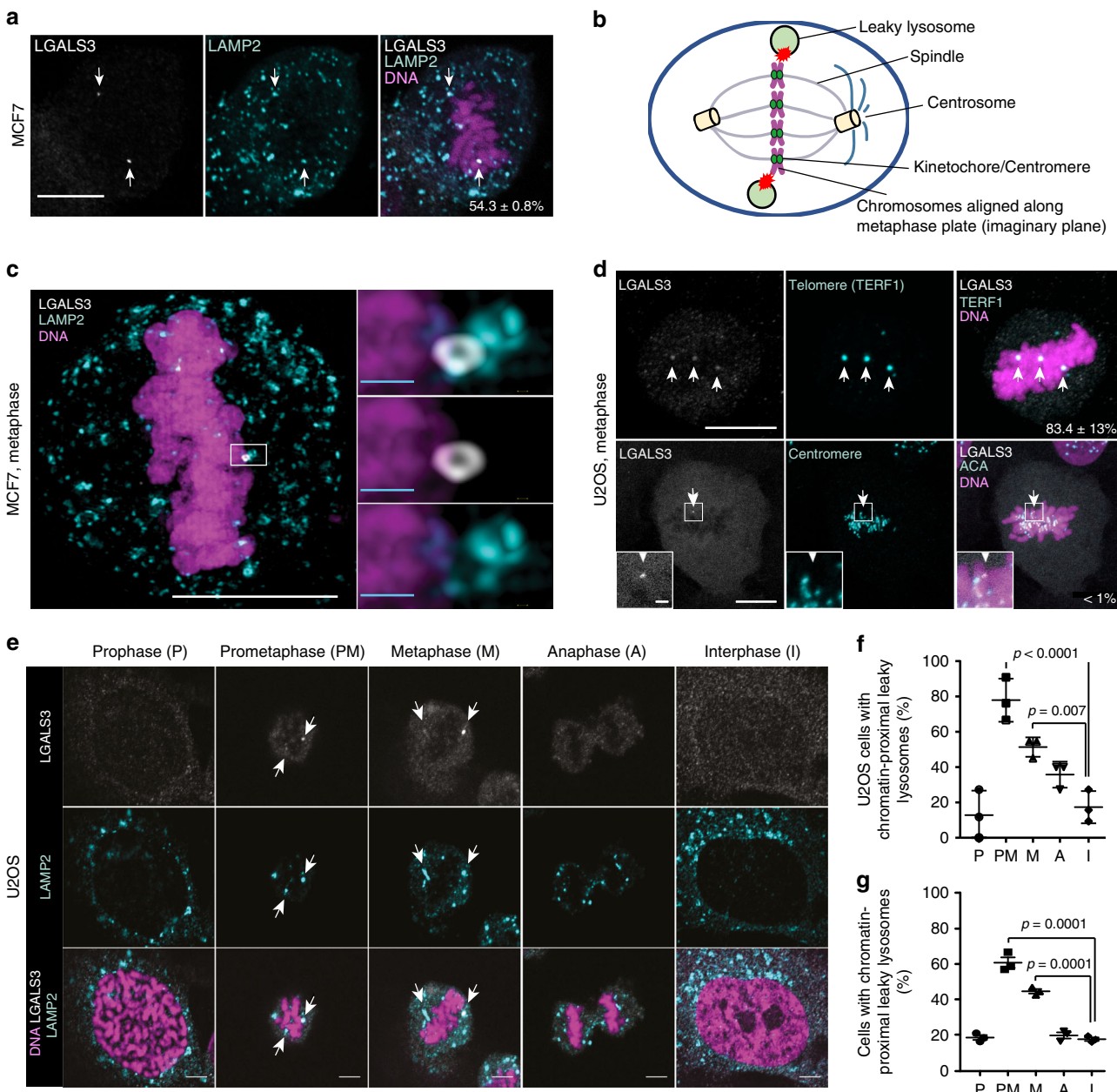

**Fig. 1 Leakage of telomere-proximal lysosomes during mitosis in vitro. a** Representative confocal images of an MCF7 cell in metaphase stained for LGALS3, LAMP1 and DNA as indicated. Value, mean percentage of metaphases with chromatin-proximal leaky lysosomes ± SD, n = 3 independent experiments with 129 metaphases analyzed. **b** Schematic presentation of a positioning of leaky lysosomes in metaphase. **c** Representative super-resolution structured illumination microscopy image of a leaky, LGALS3-positive lysosome in a mitotic MCF7 cell from 1 experiment with 10 mitoses analyzed. **d** Representative confocal images of U2OS cells stained for LGALS3, telomeres (anti-TERF1) or centromeres (anti-centromere antibody; ACA) and DNA. Values, mean percentage of chromatin-proximal LGALS3 puncta co-localizing with TERF1 or centromeres ± SD, n = 3 independent experiments with 29 randomly chosen areas with 61 metaphases analyzed. White squares mark areas magnified in lower left corners. **e** Representative confocal images of leaky lysosomes in synchronized U2OS cells in indicated phases of cell cycle (n > 50 for each phase). See Supplementary Fig. 1e for the synchronization protocol. **f** Quantification of (**e**). Dot plots, mean ± SD, n = 3 independent experiments with ≥ 10 cells analyzed for each phase in each sample. **g** Quantification of leaky lysosomes in synchronized MCF7 cells in indicated phases of cell cycle. Dot plots, mean ± SD of three independent experiments with ≥22 cells analyzed for each phase in each sample. P-values were calculated by one-way ANOVA combined with Dunnett's multiple comparisons. Arrows, chromatin-proximal leaky lysosomes. Scale bars, white 10 µm; yellow 1µm; light blue 0,5 µm. Source data for **a**, **d**, **f**, **g** are provided as Source Data files.

lysosomes were found in the proximity of kinetochores in the centromeric regions of the mitotic chromosomes (Fig. 1d and Supplementary Fig. 1b).

As discussed above, lysosomal leakage can trigger the intrinsic apoptosis pathway via cysteine cathepsin-mediated cleavage of various BCL2 family proteins[42]. Consistent with the high viability of U2OS cells in the optimal culture conditions used here, mitotic cells containing LGALS3-positive lysosomes were, however, devoid of BAX activation (Supplementary Fig. 1c), a sensitive marker for apoptosis[43], and other signs of cell death. Lysophagy, a targeted autophagy of lysosomes, has recently emerged as the mechanism by which cells can remove damaged lysosomes[44]. In line with a dramatic decline in the autophagic activity during mitosis[45], the majority of leaky mitotic lysosomes did not co-

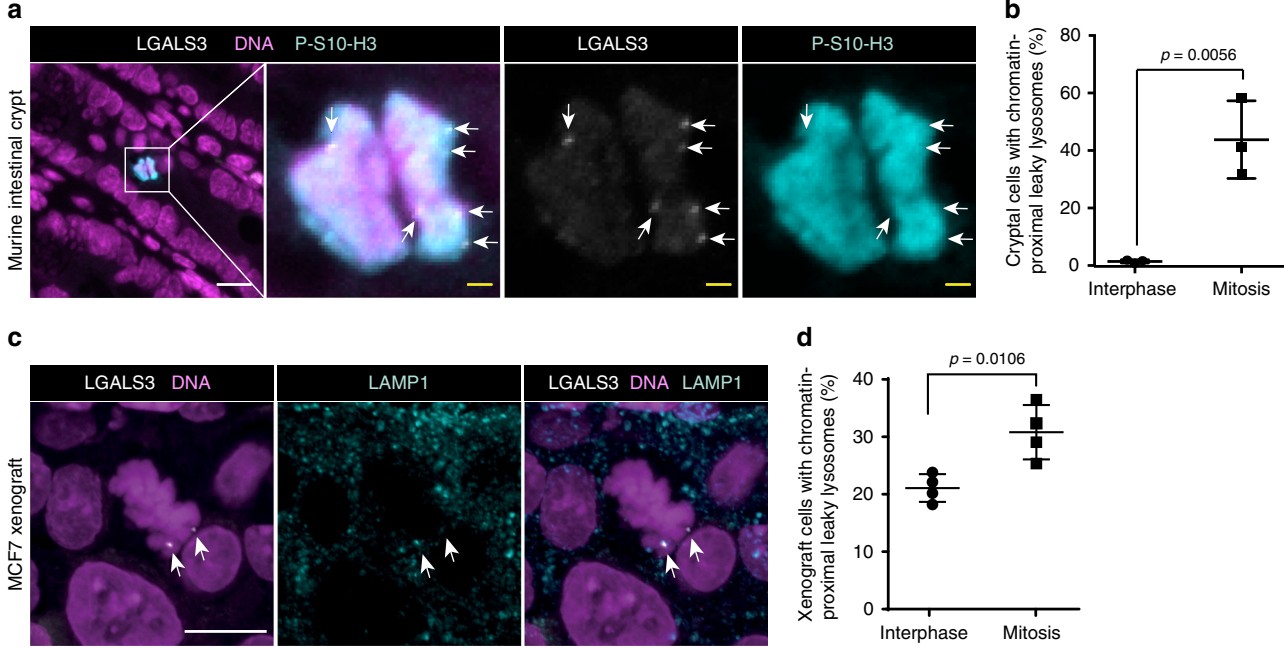

**Fig. 2 Leakage of chromatin-proximal lysosomes during mitosis in vivo. a** Representative confocal images of chromatin-proximal leaky lysosomes in murine intestinal crypts stained for LGALS3, mitotic chromatin (P-S10-H3) and DNA. **b** Quantification of (**a**). Dot plots, mean ± SD, $n = 3$ mice with ≥324 mitotic and ≥356 interphase cells from 3 non-consecutive sections analyzed for each mouse. See Supplementary Fig. 2a for the co-localization of LGALS3 and LAMP1 in intestinal crypts. **c** Representative confocal images of chromatin-proximal leaky lysosomes in orthotopic MCF7 breast carcinoma xenografts from Balb/c *nude* mice stained for LGALS3, LAMP1, and DNA. (d) Quantification of (**c**). Dot plots, mean ± SD, $n = 4$ mice with ≥165 mitotic and ≥156 interphase cells from 3 non-consecutive sections analyzed for each mouse. Arrows, chromatin-proximal leaky lysosomes. *P*-values were calculated by unpaired, two-tailed Students t-test. Scale bars, 10 μm. Source data for **b**, **d** are provided as Source Data files.

localize with EGFP-LC3 positive autophagic membranes (Supplementary Fig. 1d). Instead, the SR-SIM images suggest that they may recover by fusing with LGALS3-negative healthy lysosomes (Fig. 1c).

To define the stage of mitosis, in which lysosomes leak, we synchronized U2OS and MCF7 cells in the late G2 phase of the cell cycle by treating them with a CDK1 inhibitor (RO3306) for 16 h, released them from the G2 block and followed the fate of LGALS3-positive lysosomes throughout the mitosis (Supplementary Fig. 1e). Percentage of cells with chromatin-proximal LGALS3-positive lysosomes remained at low interphase level in prophase, peaked in prometaphase and declined thereafter close to interphase values in anaphase, when LGALS3-positive lysosomes either moved to the cell periphery and distributed to daughter cells or disappeared (Fig. 1e–g; Supplementary Fig. 1f, g and Supplementary Movie 1). The mitosis-specific lysosomal leakage was not limited to cultured cancer cells but was observed also in non-transformed human HEK293 epithelial kidney cells (Supplementary Fig. 1h), cryptal cells in healthy murine intestines (Fig. 2a, b; Supplementary Fig. 2a), and tumor cells in MCF7 breast cancer xenografts in mice (Fig. 2c, d). Leaky lysosomes were also frequently observed in random mitotic interstitial cells in non-dividing murine tissues, such as kidney (Supplementary Fig. 2b).

Together, these data reveal a non-lethal lysosomal leakage, which occurs in the close proximity of telomeres shortly after the transition from prophase to prometaphase, as a widespread phenomenon not only in cultured mammalian cells but also in healthy and cancerous tissues in vivo.

**Inhibition of lysosomal leakage causes segregation errors.** Fenton reaction and lipid composition of lysosomal membranes are among the few known regulators of lysosomal membrane

integrity[13]. Hydroxyl radicals and oxidized iron generated by Fenton reaction can lead to lipid peroxidation and subsequent destabilization of lysosomal membranes[20], whereas excess cholesterol and acid sphingomyelinase activity promote lysosomal stability[24,26,46]. To enlighten the mechanisms controlling mitotic lysosomal leakage, we treated cells with an iron chelator (deferiprone), a lipid peroxidation inhibitor (α-tocopherol), antioxidants (trolox and buthionine sulfoximine), cholesterol, a cholesterol biosynthesis inhibitor (simvastatin) or an activator of lysosomal acid sphingomyelinase (recombinant heat shock protein 70; rHSP70)[24], and quantified chromatin-proximal leaky lysosomes in metaphase. Short-term loading of lysosomes with cholesterol during mitotic entry reduced the chromatin-proximal lysosomal leakage significantly without affecting the lysosomal cysteine cathepsin activity (Fig. 3a, b; Supplementary Fig. 3a–d). The cholesterol-induced increase in lysosomal membrane stability was associated with a significant increase in erroneous chromosome segregation, mainly manifesting as anaphase bridges, i.e. physical links between incompletely separated sister chromatids or joined chromatids or chromosomes, in the subsequent anaphase (Fig. 3c, d; Supplementary Fig. 3c). Contrary to cholesterol, the other tested treatments affected neither the lysosomal leakage in metaphase nor the chromosome segregation in anaphase (Fig. 3a–d; Supplementary Fig. 3e, f). These data suggest that the mitotic chromatin-proximal leakage of lysosomes may promote the accurate segregation of chromosomes.

How could lysosomal leakage be linked to chromosome segregation? One possibility is that extra-lysosomal hydrolases assist chromosome segregation by degrading key substrates on the chromatin. Supporting this, *CTSB* itself as well as *M6PR* (mannose-6 phosphate receptor) that is essential for the lysosomal localization of CTSB and other lysosomal hydrolases[1], RAB29 that is responsible for recycling M6PR back to Golgi apparatus[47], and *MZF1* (myeloid zinc finger 1) transcription

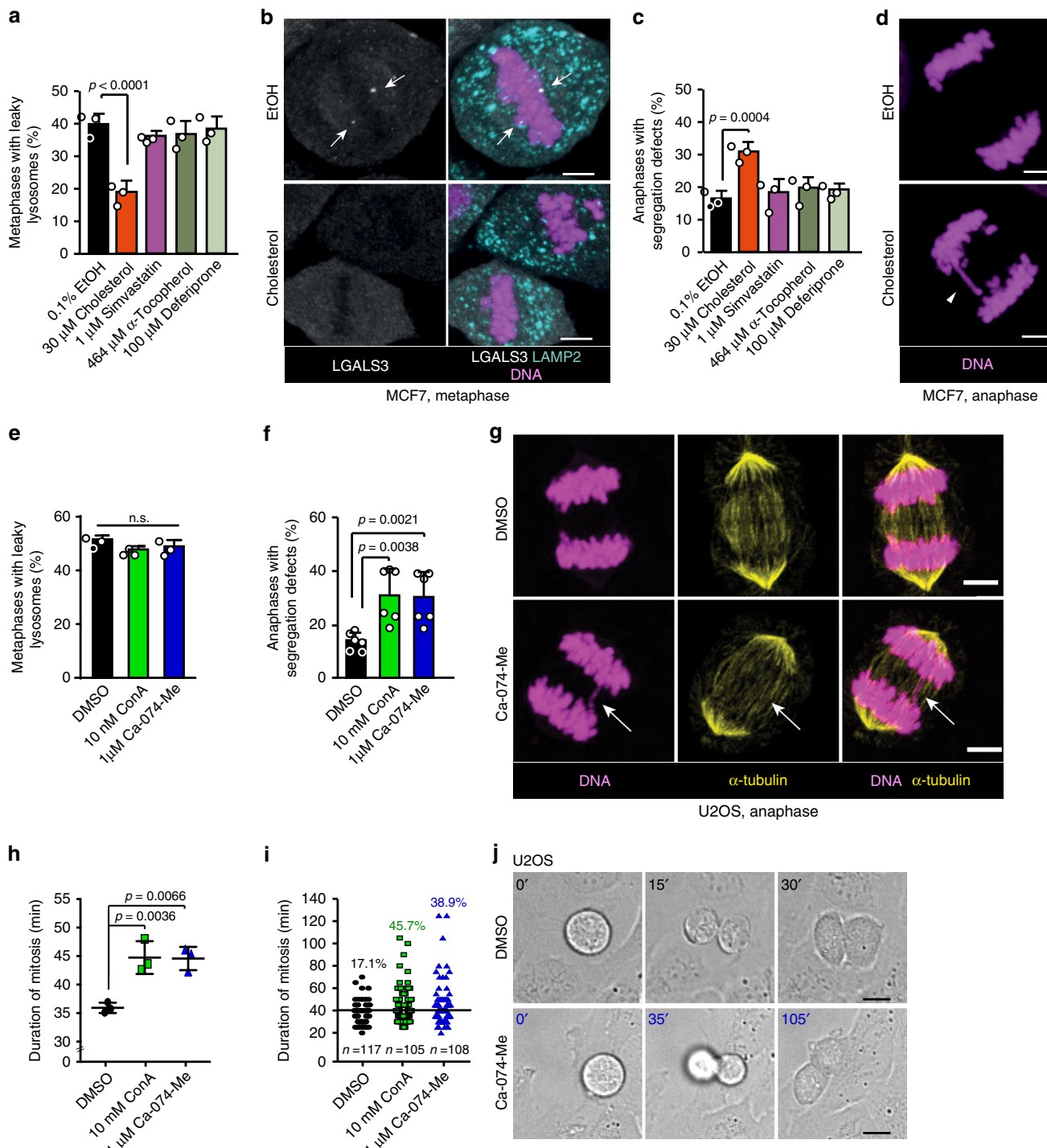

**Fig. 3 Inhibition of lysosomal leakage or activity during mitotic entry causes chromosome segregation errors. a** Quantification of metaphases with chromatin-proximal leaky lysosomes in MCF7 cells treated as indicated for 2 h upon release from late G2 arrest. Bars, mean + SD, $n = 3$ independent experiments with ≥50 metaphases analyzed for each sample. **b** Representative confocal images of (**a**). Arrows, leaky lysosomes. Scale bars, 5 μm. **c** Quantification of chromosome segregation errors in MCF7 cells treated as in (**a**). Bars, mean + SD, $n = 3$ independent experiments with ≥50 anaphases analyzed for each sample. **d** Representative confocal images of (**c**). Arrowhead, anaphase bridge. Scale bars, 5 μm. **e** Quantification of metaphases with chromatin-proximal leaky lysosomes in U2OS cells treated as indicated for 1.5 h upon release from late G2 arrest. Bars, mean + SD, $n = 3$ independent experiments with >70 metaphases analyzed for each sample. **f** Quantification of chromosome segregation errors in U2OS cells treated as indicated for 1.5 h upon release from late G2 arrest. Bars, mean + SD, $n = 6$ independent experiments with ≥50 anaphases analyzed for each sample. **g** Representative confocal images of (**f**). Arrow, anaphase bridge. Scale bars, 5 μm. **h** Duration of mitosis in non-synchronized U2OS cells that entered mitosis 1–6 h after indicated treatments. Dot plot, mean duration of mitosis ± SD, $n = 3$ independent experiments with ≥10 mitoses analyzed for each sample. **i** Scatter plot of (**h**). The percentages of mitoses lasting >40 min are indicated. **j** Representative bright-field images of (**h**). The minutes indicate the time from the rounding of the cell. Scale bar, 10 μm. *P*-values were calculated by one-way ANOVA combined with Dunnett's multiple comparisons. Source data for **a**, **c**, **e**, **f**, **h**, **i** are provided as Source Data files.

factor that enhances *CTSB* and cathepsin L (*CTSL*) transcription[48], were among the 572 genes whose depletion resulted in severe mitotic defects in a previously published, genome-wide, image-based siRNA screen for mitosis regulators in HeLa cervix carcinoma cells (http://www.mitocheck.org)[49]. To test the validity of the published screen data, we treated cells with a direct inhibitor of CTSB (Ca-074-Me) or an inhibitor of vacuolar H$^+$-ATPase (Concanamycin A; ConA)[50], which inhibits lysosomal acidification and thereby the proteolytic activation of CTSB. A short-term treatment of cells with these inhibitors during mitotic entry had no effect on the leakiness of mitotic lysosomes in metaphase (Fig. 3e, alignment of chromosomes in metaphase or the formation of the mitotic spindle (Fig. 3g), but increased significantly the errors in chromosome segregation visible in anaphase (Fig. 3f), and delayed the mitotic progression of a subset of cells as measured from the detachment of a cell to the reattachment of the two daughter cells (Fig. 3h–i; Supplementary Movies 2–4). It should be noted that the time from nuclear envelope breakdown to the start of anaphase was not affected by ConA pretreatment (Supplementary Fig. 3g). These data are in concordance with the above-discussed screen, and suggest that CTSB is the lysosomal effector molecule that promotes the correct segregation of chromosomes in mitosis.

**CTSB depletion causes mitotic defects in vitro and in vivo.** As discussed above, errors in chromosome segregation during mitosis can cause genomic instability and aneuploidy[31,35]. In order to test whether the segregation errors observed following a short-term inhibition of CTSB activity during a single mitosis translated to additional nuclear abnormalities during prolonged CTSB deficiency, we used genetic means to deplete U2OS cells for *CTSB*. In addition to increases in mitotic cells with missegregated chromosomes, *CTSB* depletion by three independent siRNAs for 3–4 days resulted in an up to 4.3-fold increase in micronuclei-containing cells and an increase in cells in G2/M phase of the cell cycle (Fig. 4a–d; Supplementary Fig. 4a–c; Supplementary Movies 5a–c). Notably, the abundance of mitotic defects correlated with the efficacy of the tested siRNAs, and the enhanced downregulation of *CTSB* expression obtained by a double transfection resulted in a further increase in chromosome segregation errors (Fig. 4a–d; Supplementary Fig. 4a). Severe signs of genomic instability and aneuploidy, i.e. micronuclei and polynucleated cells, accumulated also in two independent clones of *CTSB* deficient U2OS cells created by CRISPR/Cas9-based gene editing (Fig. 4e, f; Supplementary Fig. 4d). In order to exclude the possibility that the observed mitotic problems were caused by cell culture artefacts, we compared mitosis-rich tissues, intestinal crypts and epidermis, from wild type mice expressing *Ctsb* in these tissues with the same tissues from *Ctsb*$^{-/-}$ mice[51]. Even though previous studies have not reported any specific phenotypes in *Ctsb*$^{-/-}$ intestines or epidermis[52], careful analyses of their nuclear morphology revealed a significant accumulation of micronuclei in *Ctsb*$^{-/-}$ tissues (Fig. 4g, h). A similar nuclear phenotype was observed in intestines and epidermis from mice deficient for both *Ctsb* and *Ctsl* (Fig. 4g, h). Akin to the normal tissues, *Ctsb*$^{-/-}$ pancreatic neuroendocrine tumors had more micronuclei than wild type tumors, whereas the deficiency of a related cysteine cathepsin, *Ctss*, had no detectable impact on the nuclear morphology (Supplementary Figure 4e). In the pancreatic tumors, the effect of *Ctsb* deficiency was, however, not statistically significant, probably due to a significantly decreased proliferation and increased cell death in *Ctsb*$^{-/-}$ tumors as compared to wild type and *Ctss*$^{-/-}$ tumors[53]. The severe mitotic defects observed in cells and murine tissues upon pharmacological or genetic inhibition of *CTSB/Ctsb* strongly support the notion that this hydrolase is required for proper chromosome dynamics during mitosis.

**CTSB inhibition causes telomere-related segregation defects.** Erroneous kinetochore-microtubule attachments can be either syntelic (both kinetochores attached to the microtubules emanating from a single spindle pole), or merotelic (one kinetochore attached to both spindle poles), the latter being the main cause of lagging chromosomes in anaphase[54]. On the other hand, problems caused by under-replicated DNA, intertwined sister chromatids or entangled telomeres in the presence of normal amphitelic kinetochore-microtubule attachments result in the formation of chromatin bridges with two centromeres/kinetochores segregating to the opposite poles[36,38]. In order to enlighten the role of CTSB in chromosome segregation, we characterized the types of segregation errors caused by lysosomal inhibitors by combining immunolabelling of kinetochores/centromeres with detection of telomeres by fluorescence in situ hybridization (FISH). In addition to the expected, low fraction of lagging chromosomes and acentric chromosome fragments in U2OS cells, the elevated levels of chromosome segregation errors caused by ConA and Ca-074-Me were mainly dicentric bridges with central telomeres and peripheral kinetochores/centromeres (Fig. 5a–c; Supplementary Fig. 5a), and thus not a consequence of erroneous kinetochore-microtubule attachments, but more likely caused by entangled telomeres. Normal alignment of metaphase chromosomes as well as unchanged kinetochore levels of total and active Aurora B kinase, which is responsible for the correction of most erroneous kinetochore-microtubule attachments[55], further supported a kinetochore-unrelated mechanism (Supplementary Fig. 5b). In line with this, the majority of the accumulating segregation defects caused by the treatment of cells with *CTSB* siRNA for 3 days were either dicentric chromatin bridges with amphitelic kinetochore-microtubule attachments and central telomeres or telomere-positive acentric fragments devoid of centromeres/kinetochores (Fig. 5d). These data suggest that CTSB assists in untangling fused telomeres to promote accurate chromosome segregation.

**Telomere dysfunction increases mitotic lysosomal leakage.** Based on the localization of leaky lysosomes in the close proximity of telomeres (Fig. 1d), appearance of leaky lysosomes in prometaphase shortly after the nuclear membrane disintegrates (Fig. 1e–g), and the abundance of telomere-related segregation defects upon CTSB inhibition (Fig. 5c, d), we speculated that entangled telomeres or factors associated with them could serve as triggers for the lysosomal leakage. To test this hypothesis, we depleted cells for telomeric repeat binding factor 2 (*TERF2*), which is responsible for the formation of the telomere protecting t-loop structure and whose depletion thereby induces an excessive formation of telomere fusions[36,56]. *TERF2* depletion caused a significant increase in both the percentage of (pro)metaphases with chromatin-proximal, leaky lysosomes and the number of chromatin-proximal, leaky lysosomes *per* (pro)metaphase already 48 h after the addition of the siRNA (Fig. 5e–i). These data provide indirect support for the idea that entangled telomeres may serve as a trigger for the lysosomal leakage upon nuclear membrane disintegration in (pro)metaphase. Further studies are, however, required to identify detailed molecular mechanisms governing this process.

**Active CTSB decorates metaphase chromatin.** The data presented above suggest that CTSB leaks out of the lysosomal compartment in prometaphase to assist chromosome segregation later in mitosis. To define whether CTSB actually associates with

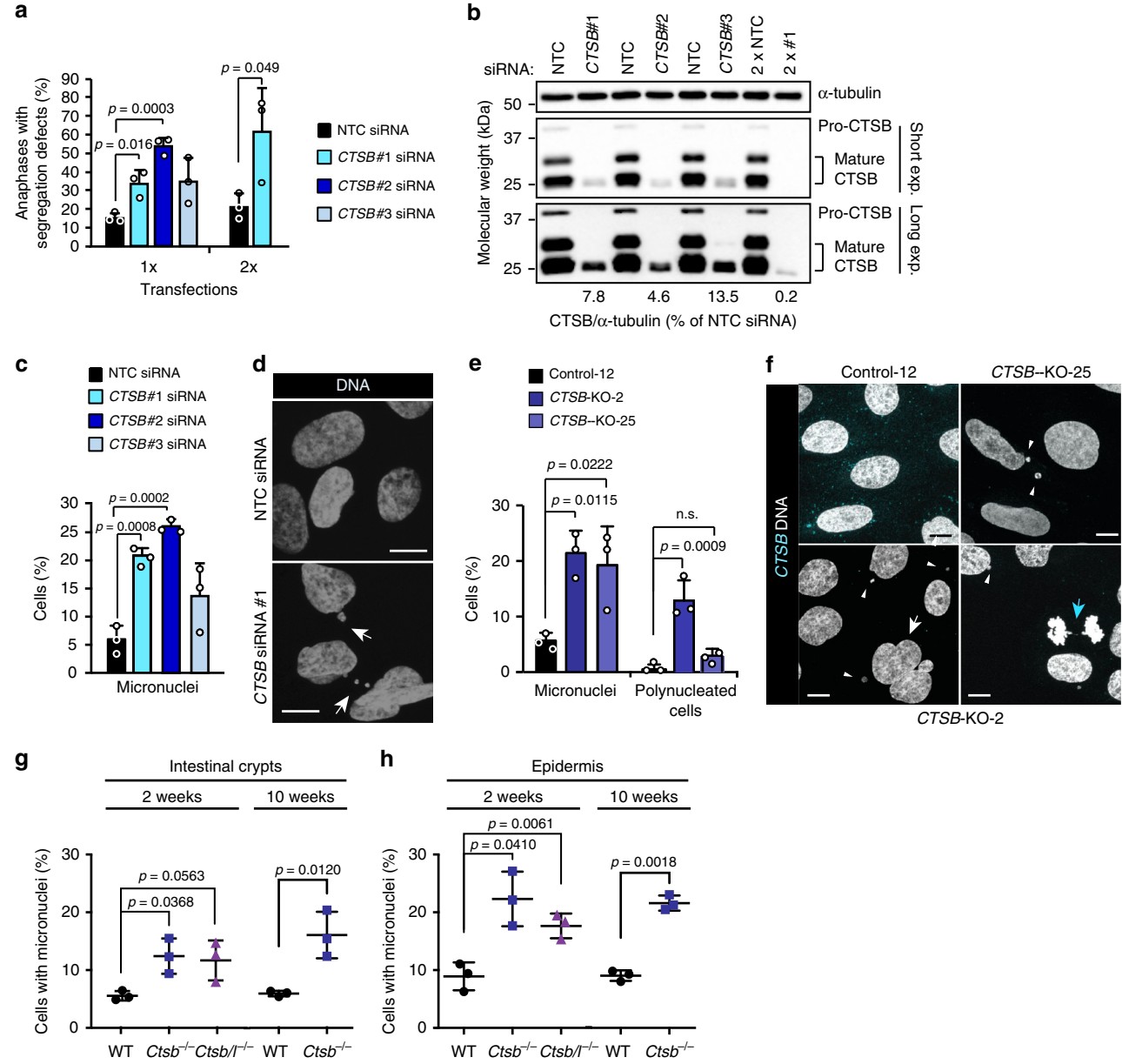

**Fig. 4 Depletion of *CTSB* causes mitotic defects and nuclear abnormalities. a** Quantification of anaphases with segregation defects in U2OS-H2B-GFP cells transfected with indicated siRNAs once for 72 h (1×) or twice for 48 h (2×). Bars, mean + SD, n = 3 independent experiments with ≥22 anaphases analyzed for each sample. See Supplementary Fig. 4c and Supplementary Movie 5 for images of anaphase defects. **b** Representative immunoblots of cells used in (**a**), n = 3 independent experiments. **c** Quantification of interphases with micronuclei in U2OS cells transfected with indicated siRNAs for 72 h. Bars, mean + SD, n = 3 independent experiments with >165 interphases analyzed for each sample. See Supplementary Fig. 4a for CTSB immunoblots. **d** Representative confocal images of (**c**). Arrows, micronuclei. **e** Quantification of mitotic defects in indicted U2OS CRISPR clones stained for CTSB and DNA. Bars, mean + SD, n = 3 independent experiments with >200 cells analyzed for each sample. **f** Representative confocal images of (**e**). Arrowheads, micronuclei; white arrow, polynucleated cell; blue arrow, chromatin bridge. See Supplementary Fig. 4d for CTSB immunoblots. **g** Quantification of cells with micronuclei in intestinal crypts from indicated mouse strains at the age of 2 or 10 weeks. Dot blots, mean ± SD, n = 3 mice with ≥867 (2 weeks) or ≥586 (10 weeks) cells from three non-consecutive sections analyzed for each mouse. **h** Quantification of cells with micronuclei in epidermis from indicated mouse strains at the age of 2 or 10 weeks. Dot blots, mean ± SD, n = 3 mice with ≥555 (2 weeks) or ≥160 (10 weeks) cells from ≥3 non-consecutive sections analyzed for each mouse. P-values were calculated by unpaired, two-tailed Students t-test (**a**) or one-way ANOVA combined with Dunnett's multiple comparisons (**b**, **c**). Scale bars, 10 µm. Source data for panels **a**, **b**, **c**, **e**, **g**, **h** are provided as Source Data files.

the mitotic chromatin, we set up a proximity ligation assay (PLA) employing antibodies against CTSB and Ser-10-phosphorylated Histone H3 (P-S10-H3), a marker of mitotic chromatin. Approximately 75% of vehicle-treated cells had an abundant mitosis-specific PLA signal (Fig. 6a, b), which was practically absent in *CTSB* deficient cells (Supplementary Fig. 6a). Short treatments with either ConA or Ca-074-Me upon mitotic entry,

which did not prevent the leakage of lysosomes in (pro)metaphase but increased chromosome segregation errors in anaphase (Fig. 3e–g), reduced the percentage of PLA-positive cells significantly (Fig. 6a, b). Because ConA and Ca-074-Me inhibit the binding of CTSB to its substrates by inhibiting the proteolytic maturation of CTSB and by binding to the active site of CTSB[57], respectively, the lack of the PLA signal following these treatments

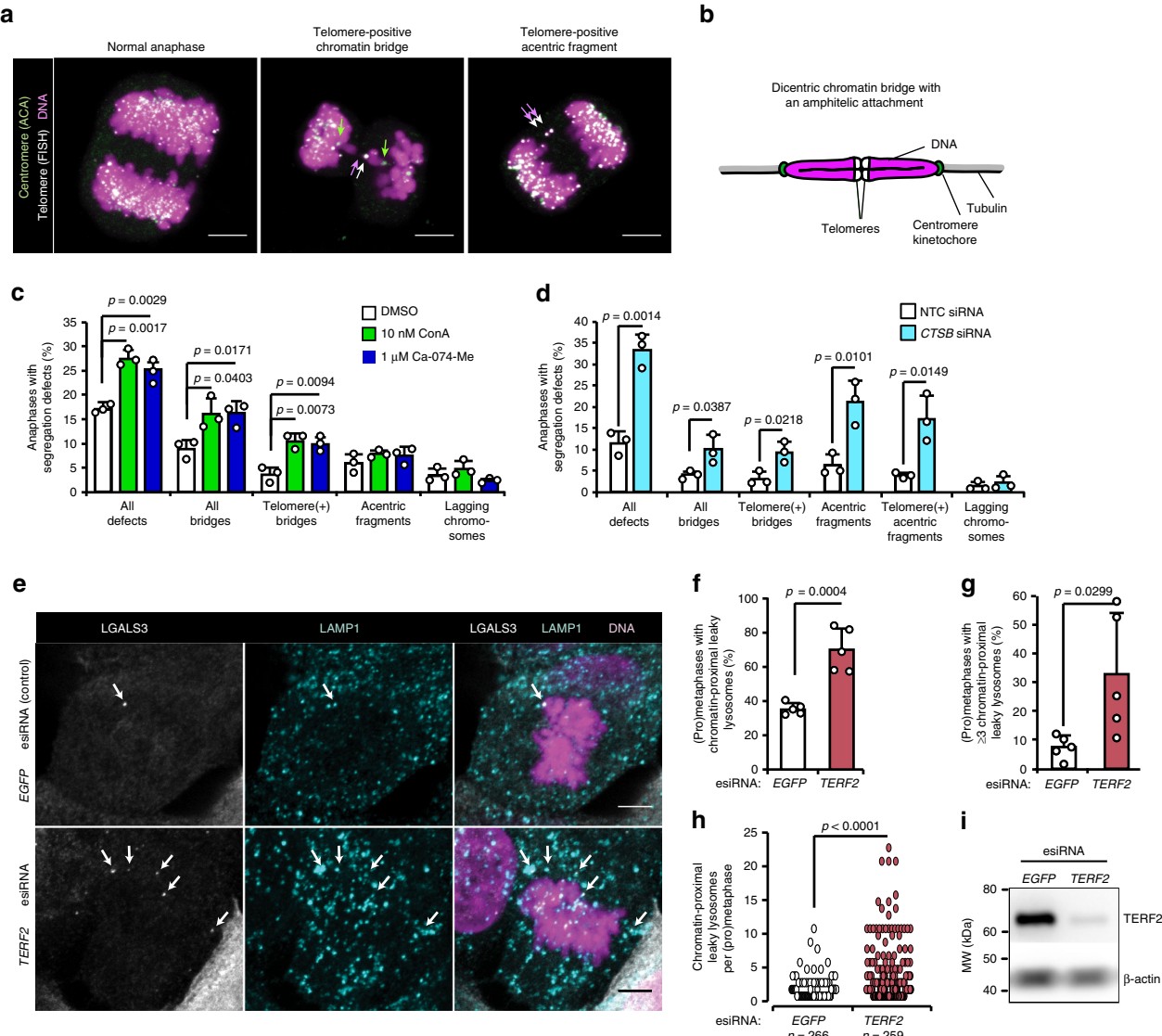

**Fig. 5 Inhibition of CTSB causes telomere-related chromosome segregation defects. a** Representative confocal images of a normal anaphase and anaphases with indicated defects in U2OS cells treated with DMSO (normal anaphase) or 10 nM ConA for 1.5 h upon release from late G2 arrest and stained as indicated, $n = 3$ independent experiments. White, green, and magenta arrows mark telomeres, centromeres/kinetochores, and DNA in chromosome segregation defects, respectively. See Supplementary Fig. 5a for images with single colors. **b** Schematic presentation of a telomere-positive chromatin bridge with an amphitelic attachment. **c** Quantification of indicated chromosome segregation defects in U2OS cells treated as indicated for 1.5 h upon release from late G2 arrest. Bars, mean + SD, $n = 3$ independent experiments with >160 anaphases analyzed for each sample. **d** Quantification of indicated chromosome segregation defects in U2OS cells transfected with indicated siRNAs for 48 h. Bars, mean + SD, $n = 3$ independent experiments with ≥70 anaphases analyzed for each sample. **e** Representative confocal images of chromatin-proximal leaky lysosomes in (pro)metaphases of U2OS cells transfected with EGFP (control) or TERF2 esiRNA for 48 h. White arrows mark leaky lysosomes. **f**–**h** Quantifications of (**e**). Bars, mean + SD, $n = 5$ independent experiments with ≥40 (pro)metaphases analyzed for each sample (**f**, **g**) and distribution of leaky lysosomes in all analyzed cells (**h**). **i** Representative immunoblots showing the efficacy of TERF2 esiRNA, $n = 3$ independent experiments. Scale bars, 5 μm. *P*-values were calculated by unpaired, two-tailed Students *t*-test. Source data for **c**, **d**, **f**, **g**, **h**, **i** are provided as Source Data files.

suggests that CTSB has to be in its active, substrate-binding form in order to associate with chromatin. The association of active CTSB with metaphase chromatin was further reinforced by the close proximity of CTSB and double-stranded DNA (dsDNA) in all studied metaphase chromosome spreads from control U2OS cells and its absence in similar chromosome spreads from Ca-074-Me-treated cells (Supplementary Fig. 6b). Accordingly, staining of the metaphase cells with a lysosomal marker (AlexaFluor® 488-Dextran) and Magic Red, which becomes fluorescent upon CTSB-mediated cleavage, revealed extra-lysosomal, chromatin-proximal CTSB activity in 80% of

normal metaphases (Fig. 6c, d). Furthermore, immunoblotting of lysates of metaphase-enriched control cells for histone H3, a previously identified substrate of CTSB and CTSL[12,58], revealed a minor histone H3 band with increased motility, which was absent in similar lysates from cells treated with Ca-074-Me or ConA upon mitotic entry (Fig. 6e). Together, these data strongly support the idea that active CTSB associates with metaphase chromatin.

**Histone H3 cleavage by CTSB promotes chromosome segregation.** To investigate whether the chromatin-associated cathepsin

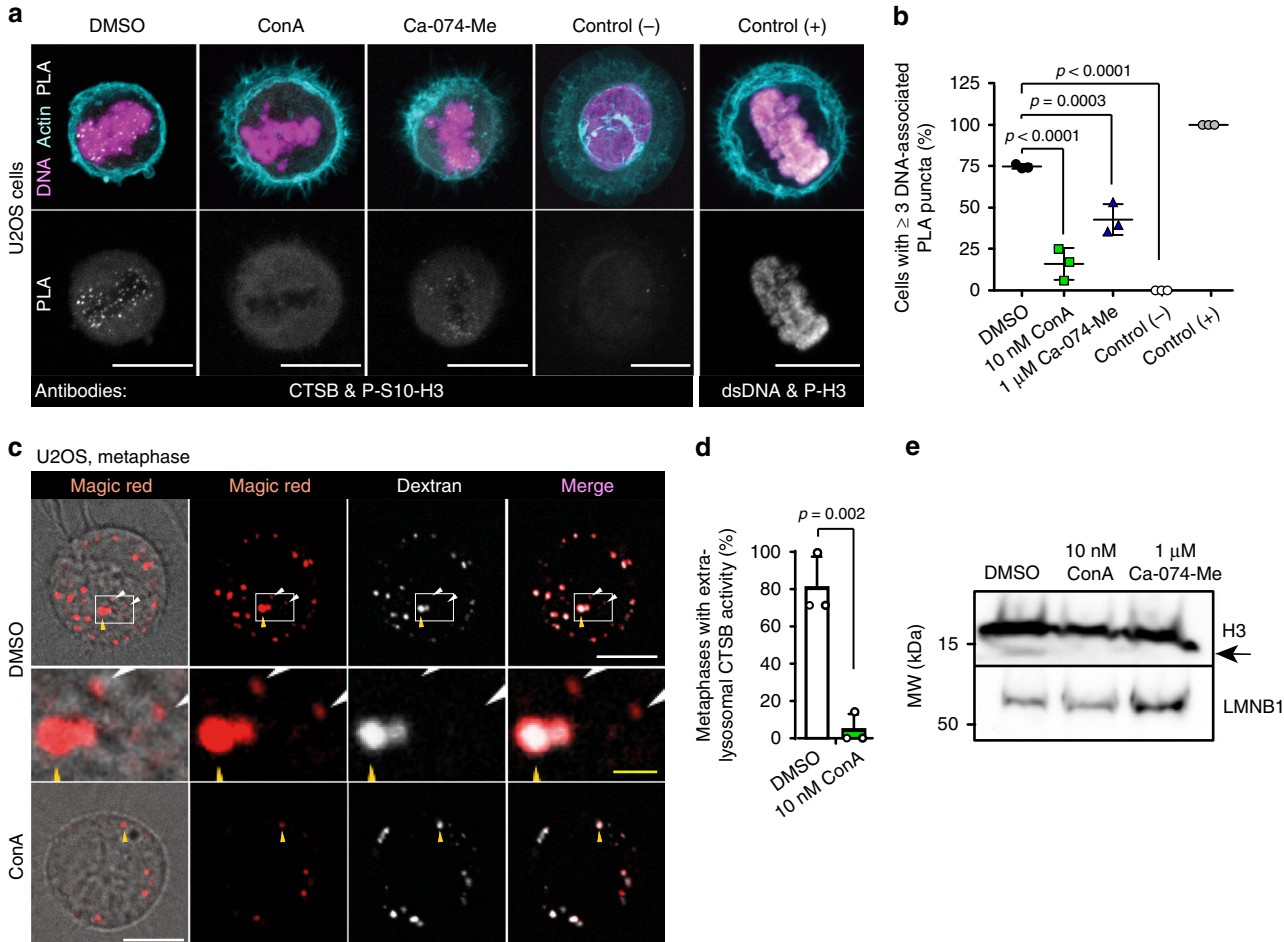

**Fig. 6 Active CTSB decorates metaphase chromatin. a** Representative confocal images of PLA puncta with indicated antibodies in U2OS cells treated as indicated for 1.5 h upon release from late G2 arrest. Interphase cells (Control (−)) and CTSB-KO cells (Supplementary Fig. 6a) served as negative controls, and proximity of P-S10-H3 and dsDNA in metaphase cells as a positive control. Cells were counterstained for actin and DNA. **b** Quantification of (**a**). Dot plot, mean ± SD, $n = 3$ independent experiments with ≥15 cells analyzed for each sample. **c** Representative bright field and confocal images of metaphase U2OS cells pre-loaded with AlexaFluor® 488-Dextran (lysosomal marker) and treated with Magic Red (CTSB activity probe) and DMSO or 10 nM ConA 15 min prior to the start of 2 h recording. White arrowheads, extralysosomal CTSB activity (red dots outside white lysosomes); yellow arrowhead, lysosomal CTSB activity (red dots co-localizing with white lysosomes). **d** Quantification of (**c**). Bars, mean + SD, $n = 3$ independent experiments with ≥4 metaphases analyzed per condition. **e** Representative immunoblots of histone H3 and LMNB1 in lysates of U2OS cells treated as indicated for 1.5 h upon release from late G2 arrest, $n = 3$ independent experiments. Arrow, histone H3 cleavage fragment. *P*-values were calculated by one-way ANOVA combined with Dunnett's multiple comparisons (**a**) or unpaired, two-tailed Students *t*-test (**b**). Scale bars, 10 µm (white), 1 µM (yellow). Source data for panels b, d and e are provided as Source Data files.

activity is essential for the proper chromosome segregation, we expressed cystatin B (CSTB), a cytosolic cysteine cathepsin inhibitor, in its native cytosolic form or fused to either histone H2B or H3 in U2OS cells, and compared the number of segregation errors in these cells with cells transfected with an empty vector or EGFP fused to histones H2B or H3. All ectopic proteins were expressed in comparable levels, and as expected, wild type CSTB localized to the cytosol and effectively inhibited the cysteine cathepsin activity in cell lysates, whereas both CSTB and EGFP fused to either one of the histones co-localized with DNA (Fig. 7a, b; Supplementary Fig. 7a). Indicative of local, chromatin-associated cathepsin inhibitory activity of CSTB-histone H2B and CSTB-histone H3 fusion proteins, chromatin extracts from cells expressing them inhibited the activity of recombinant CTSL (rCTSL) in vitro by 84% and 81%, respectively, as compared to the activity in the presence of chromatin from cells expressing the corresponding EGFP-histone fusion proteins (Fig. 7c). The expression of H3-CSTB, but not that of wild type CSTB or H2B-CSTB, significantly increased the number of chromosome segregation errors

(Fig. 7d), suggesting that cysteine cathepsin-mediated cleavage of histone H3 could promote accurate chromosome segregation. In vitro cleavage assay of chromatin extracts from transfected cells confirmed that histone H3-EGFP was cleaved by both recombinant CTSB (rCTSB) and rCTSL even at pH 7.0, whereas histone H3-CSTB was protected from the cleavage (Fig. 7e). The estimated molecular weight of the carboxy-terminal H3-EGFP cleavage fragment was approximately 10 kDa smaller than the full-length protein suggesting that the cleavage occurred at the site close to the CTSB cleavage site at Tyr-100 (Y100) of Histone H3 that was recently identified using systems-level N-terminome degradomics approach[12]. In line with this, ectopic expression of Y100 mutants (Y100A, Y100F and Y100E) in U2OS cells resulted in a significant increase in mitotic defects, including segregation errors and micronuclei (Fig. 7f, g), and purified Y100A mutant reduced its cleavage by rCTSB and rCTSL in vitro (Fig. 7h; Supplementary Fig. 7b). Notably, expression of histone H3 with a previously suggested CTSL cleavage site, Leu-21[58], substituted with tryptophan (L21W) had no detectable effect on chromosome segregation

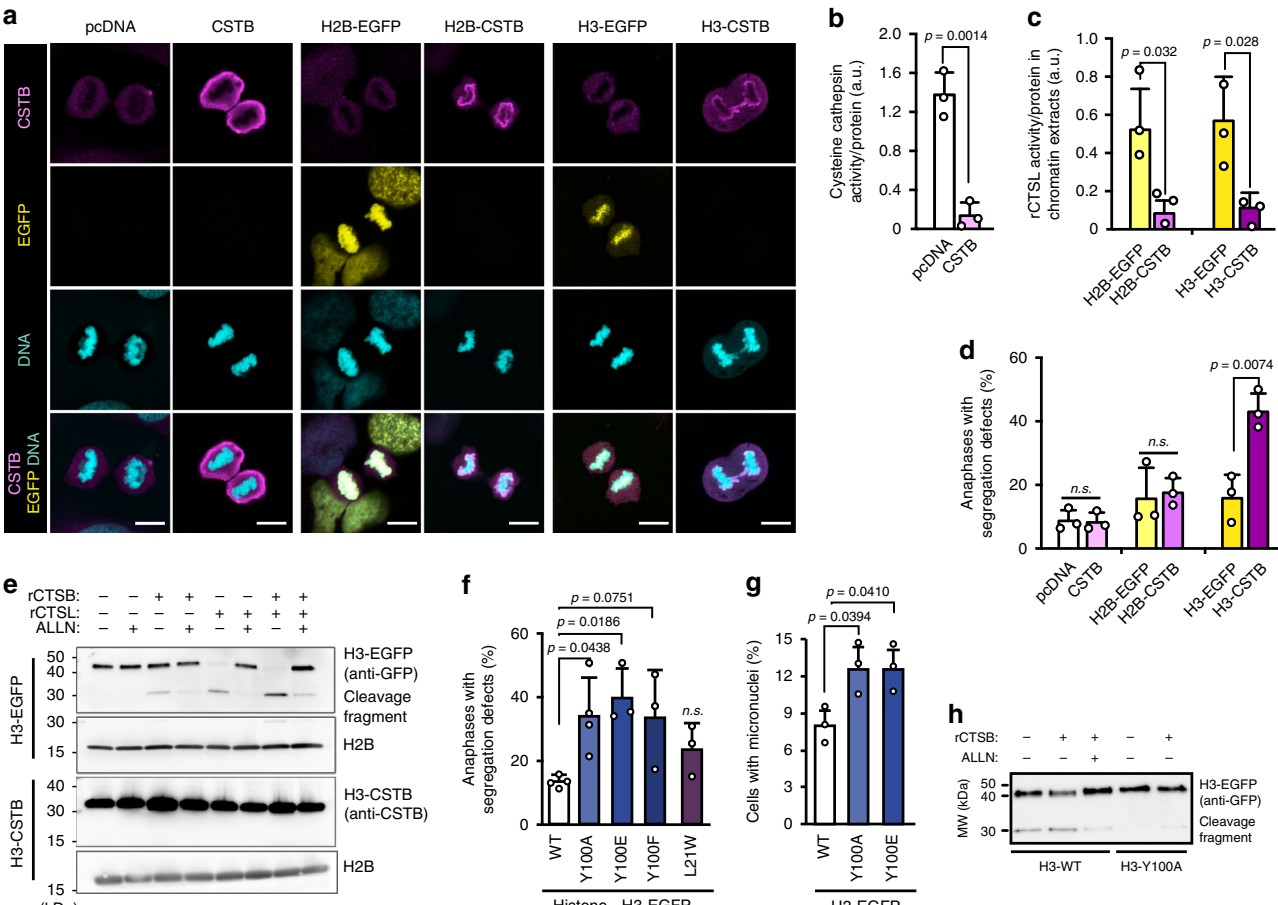

**Fig. 7 Cathepsin-mediated cleavage of histone H3 promotes chromosome segregation. a** Representative confocal images of U2OS cells transfected with indicted plasmids 48 h earlier, $n = 3$ independent experiments. See Supplementary Fig. 7a for related immunoblots. Scale bars, 10 μm. **b** Cysteine cathepsin activity (zFR-AFC cleavage) in whole cell lysates of U2OS cells transfected as in (**a**). Bars, mean + SD, $n = 3$ independent triplicate experiments. **c** Activity of rCTSL in the presence of chromatin extract from U2OS cells transfected as in (**a**) in pH 7,0. Bars, mean + SD, $n = 3$ independent triplicate experiments. **d** Quantification of anaphases with chromatin bridges in U2OS cells successfully transfected as in (**a**) 72 h earlier. Bars, mean + SD, $n = 3$ independent experiments with >50 anaphases analyzed per sample. **e** Representative immunoblots of indicated proteins in chromatin extracts of U2OS cells transfected with indicated plasmids as in (**a**) and incubated for 15 min with rCTSB, rCTSL or 200 nM ALLN at 37 °C, at pH 7.0, as indicated, $n = 3$ independent experiments. Note that H3-CSTB and H2B are from separate blots prepared in parallel due to the size overlap between a putative H3-CSTB cleavage fragment and H2B. **f** Quantification of anaphases with segregation errors in U2OS cells successfully transfected with H3-EGFP with wild type histone H3 (WT) or its indicated point mutations 72 h earlier. Bars, mean + SD, $n = 4$ (WT) or 3 (mutants) independent experiments with >50 anaphases analyzed per sample. **g** Quantification of interphases with micronuclei in U2OS cells transfected as in (**f**). Bars, mean + SD, $n = 3$ independent experiments with >100 interphases analyzed per sample. **h** Representative immunoblot of H3-EGFP and its cleavage product in chromatin extracts from U2OS cells transfected with indicated H3-EGFP constructs 48 h earlier, $n = 3$ independent experiments. Chromatin extracts were incubated for 15 min at 37 °C, at pH 7.0 with or without rCTSB and 200 nM ALLN as indicated. P-values were calculated by unpaired, two-tailed Students t-test (**b**–**d**) or one-way ANOVA combined with Dunnett's multiple comparisons (**f**, **g**). Source data for **b**–**h** are provided as Source Data files.

(Fig. 7f). In line with the ability of rCTSL to cleave histone H3 at Tyr-100, treatment of U2OS cells with CTSL inhibitor upon entry to mitosis increased the number of segregation errors in anaphase significantly, albeit to a slightly lesser extent than inhibition of CTSB by Ca-074-Me or the combined inhibition of CTSB and CTSL (Fig. 3f; Supplementary Fig. 7c). Thus, cysteine cathepsin-mediated cleavage of a subset of histone H3 in (pro)metaphase emerges as a mechanism by which lysosomal leakage promotes accurate chromosome segregation, whereas the role of other putative cathepsin substrates, as well as the role of other lysosomal hydrolases in this process, remain to be studied.

## Discussion

The lysosomal lumen is filled with a potent mixture of hydrolytic enzymes, whose leakage to the cytosol triggers caspase-dependent

apoptosis, caspase-independent apoptosis-like cell death or necrosis depending on the extent of lysosomal damage and the ability of cells to activate different cell death pathways[22,28]. Taking advantage of a highly sensitive galectin puncta assay to visualize leaky lysosomes, we show here, that a minor, non-lethal permeabilization of lysosomal membrane occurs frequently in cultured human cells and healthy murine tissues in prometaphase. Furthermore, we provide data indicating that CTSB leaking out of prometaphase lysosomes and acting on (pro)metaphase chromatin promotes faithful segregation of chromosomes in anaphase. First, stabilization of lysosomal membranes by cholesterol loading during mitotic entry results in an increase in anaphase segregation errors establishing a link between the lysosomal membrane permeabilization and accurate chromosome segregation. Second, the role of extralysosomal CTSB as an effector molecule involved in chromosome segregation is

supported by the presence of active CTSB on metaphase chromatin, increase in segregation errors upon pharmacological CTSB inhibition during mitotic entry and further accumulation of multiple nuclear defects following prolonged inhibition of CTSB by genetic means. Third, the accumulation of mitotic defects upon expression of histone H3 either mutated at the verified CTSB cleavage site or fused to a potent cysteine cathepsin inhibitor reinforces the mitotic role of chromatin-associated CTSB and introduces histone H3 as one of its substrates in (pro) metaphase. Finally, the physiological relevance of mitotic lysosomal leakage and CTSB in chromosome segregation is supported by the abundance of mitosis-specific and chromatin-proximal leaky lysosomes in healthy murine tissues and the accumulation of micronuclei in Ctsb deficient tissues.

Most chromosome segregation errors originate from problems encountered during DNA replication in the S phase of the cell division cycle[33,38]. The significant increase in chromosome segregation errors observed upon short-term inhibition of either lysosomal leakage or CTSB activity during mitotic entry, i.e. after the cells have completed the replication of their DNA, argues against a role for lysosomal leakage or extralysosomal CTSB in DNA replication. Instead, we show that CTSB associates with chromatin in (pro)metaphase, where it helps to solve segregation problems brought about by CTSB-independent mechanisms prior to mitosis, possibly during the preceding S phase. The characterization of chromosome segregation errors in the absence of CTSB activity revealed an increase in dicentric chromatin bridges with central telomeres, with no effect on the abundance of telomere-negative chromatin bridges or segregation defects associated with erroneous kinetochore-microtubule attachments, e.g., lagging chromosomes or chromatin bridges with syntelic or merotelic attachments[54]. Also upon prolonged inhibition of CTSB by RNAi, the increase in chromosome segregation errors was limited to telomere-related problems, including telomere-positive acentric fragments and dicentric chromatin bridges. Telomere-positive acentric fragments are likely to be derived from broken telomere-positive chromatin bridges and may explain the accumulation of micronuclei observed in cells and tissues lacking CTSB/Ctsb for a longer time. The chromosome segregation errors brought about by CTSB inhibition being exclusively telomere-related, it is interesting that further disturbance of telomere segregation by depletion of TERF2, a shelterin complex protein that protects telomere ends against fusion[36,56], leads to a dramatic increase in lysosomal leakage in (pro)metaphase. Together with the data showing that the leaky lysosomes appear in close proximity of telomeres, these data support a model where fused telomeres get into the contact with perinuclear lysosomes upon the disintegration of the nuclear envelope and trigger their leakage by a yet unknown mechanism. Subsequently, CTSB leaking out of the lysosomes co-localizes with the chromatin where it may help to resolve telomere-related errors, such as end-to-end fusions between sister chromatids. CTSB-mediated cleavage of a subset of histone-H3 appears to be a part of this process. CTSB being a potent hydrolase possessing both endopeptidase and exopeptidase activity even at neutral pH outside the lysosomal lumen[11,12], it is, however, likely to have additional substrates in metaphase, whose identity and role in chromosome segregation remains to studied.

Errors in chromosome segregation are common in most cancers. Even though they can cause genomic instability and thereby enable transforming cells to acquire essential cancer-promoting capabilities, chromosome segregation errors as such are not necessarily tumorigenic[31]. Instead, high rate of chromosome missegregation is more likely to promote senescence and cell death[31,59,60]. In line with this, Ctsb deficiency is not associated with a tumor phenotype in mice. Instead, it delays tumor onset and progression in various murine tumor models, i.e. MMTV-PyMT breast cancer[61], Rip1-Tag2 pancreatic neuroendocrine tumors[53], Ras$^{G12D}$/TP53$^{-/-}$ pancreatic ductal adenocarcinoma[62], and APCmin intestinal polyps[63]. The tumor suppressive role of Ctsb deficiency is, however, not solely caused by the mitotic problems described here as the frequently observed increase in Ctsb/CTSB expression in murine and human tumors stimulates tumor progression by numerous means, e.g., by supporting the increased metabolic needs of cancer cells, promoting cancer cell migration and invasion and stimulating angiogenesis[10,48]. Furthermore, extracellular Ctsb/CTSB produced mainly by tumor-infiltrating macrophages enhances cancer cell proliferation and metastasis in a paracrine fashion[10]. Thus, the loss of potent tumor-promoting functions of Ctsb may counteract the survival of a few potentially tumorigenic cells arising as a result of chromosome segregation errors in Ctsb deficient mice. In agreement with this, Ctsb deficient Rip1-Tag2 pancreatic neuroendocrine tumors have a significant increase in apoptotic cells[53]. Contrary to Ctsb deficiency, which counteracts tumor growth by multiple mechanisms, other means to interfere with the mitotic, chromatin-associated CTSB activity may favor the survival of cells with genomic instability and thereby promote tumorigenesis. In this context, it is interesting to note that several epidemiological and experimental studies have suggested a cancer-promoting role for cholesterol[64], which effectively inhibits lysosomal leakage in (pro)metaphase and increases chromosome segregation errors in anaphase in vitro. It remains, however, to be studied whether cholesterol accumulation in vivo is sufficient to stabilize lysosomes and induce chromosome segregation errors. It will also be interesting to investigate whether mitotic lysosomal leakage and the ability of CTSB to assist telomere segregation is reduced by the aging-associated general decline in lysosomal activity[65]. Such an association is indirectly supported by a recent study reporting an age-related increase in chromosomes with sister telomere loss or sister telomere chromatid fusions in peripheral blood mononuclear cells throughout the human lifespan[66].

In conclusion, we have here identified spatially and temporally defined intracellular lysosomal leakage and subsequent CTSB-mediated cleavage of a subset of histone H3 as frequent events that contribute to the appropriate segregation of telomeres in cultured human cells and healthy murine tissues. In addition to the importance of mitotic lysosomal leakage in the maintenance of genomic integrity, our data opens the possibility that controlled release of lysosomal hydrolases may also regulate other cellular processes.

## Methods

**Cell culture and treatments**. U2OS human osteosarcoma (ATCC® HTB-96™) and HEK293 human embryonic kidney (ATCC® CRL-1573™) cells were obtained from American Type Culture Collection (ATCC). The cells were authenticated by ATCC and used within 6 months after thawing. U2OS-mCherry/LGALS3 cells were kindly provided by Dr. Harald Wodrich (Laboratoire de Microbiologie Fondamentale et Pathogénicité, Bordeaux, France)[67]. U2OS-H2B–GFP cells[68], the S1 subclone of MCF7 human breast adenocarcinoma cells[69] and MCF7-EGFP-LC3 cells[70] originated from our laboratories. U2OS and MCF7 cells were authenticated by chromosome spreads and RNASeq, respectively. Cells were grown in high-glucose Dulbecco's modified Eagle's medium (DMEM, Thermo Fisher Scientific, 31966-021) supplemented with 10% fetal calf serum (FCS, Life Technologies, 10270-106) and penicillin/streptomycin (Life Technologies,15140-122). All cell lines were regularly tested and found negative for mycoplasma using Venor®GeM Classic PCR kit (Minerva Biolabs, 11-1100).

When indicated, cells were treated with indicated concentrations of α-Tocopherol (Sigma-Aldrich, T3251), Buthionine-sulfoximine (Sigma-Aldrich, B2515), Ca-074-Me (Merck Millipore, 205531), Cathepsin L inhibitor IV (Calbiochem, 219433), cholesterol (Sigma-Aldrich, C3045), concanamycin A (Santa Cruz Biotechnology, sc-202111), deferiprone (Sigma-Aldrich, 379409), dimethyl sulphoxide (DMSO, VWR Life Science, AMREN 182), recombinant Hsp70 (Ref. [24], kindly provided by Thomas Kirkegaard, Orphazyme A/S,

Copenhagen, Denmark), RO3306 (Merck Millipore, 217721), trolox (Calbiochem, 648471) or simvastatin (Sigma-Aldrich, 38956).

**Murine tissues**. MCF7 tumor xenografts were collected from control mice during a previous study[39]. Briefly, MCF7 cells ($10 \times 10^6$ cells in Geltrex (Life Technologies) in 120 µl) were inoculated into an axillary mammary fat pad of 8–10 weeks old female FOX CHASE severe combined immunodeficient (SCID/FOX, Charles River Laboratories) mice treated with 0.7 µg per mL estrone (Sigma-Aldrich, E9750) in drinking water starting 5–7 days before tumor inoculation. When tumor diameter reached 5–6 mm, animals were treated *p.o.* with 200 µl vehicle (0.5% methyl cellulose with 15 cP viscosity (Sigma-Aldrich, M7140) in 0.9% NaCl solution). After two days, mice were sacrificed, and tumors were placed in 10% formalin fixative until embedding in paraffin. The study was approved by Dyreførsøgstilsynet, Denmark.

Formalin-fixed and paraffin-embedded kidneys were from untreated C57Bl/6 mice (Charles River Laboratories) collected during a previous study[71], and kindly provided by Dr. Andreas Linkermann (Christian-Albrechts-University, Kiel, Germany). The study was approved by Christian-Albrechts-University Institutional Review Board.

Small intestines collected from 20 to 25 weeks old female FVB/N mice (Charles River Laboratories), as well as jejunum and epidermis from 2 to 10 weeks old male or female wild type C57Bl/6 or FVB/N mice, Ctsb$^{-/-}$ C57Bl/6 mice[51] and Ctsb$^{-/-}$/Ctsl$^{-/-}$ FVB/N mice[72] were placed in 10% formalin fixative and embedded in paraffin. The study was approved by the regional board Freiburg.

Pancreatic islet tumors from RIP1-Tag2, RIP1-Tag2-Ctsb$^{-/-}$ and RIP1-Tag2-Ctss$^{-/-}$ C57Bl/6 mice[53] were snap-frozen during previous studies approved by the animal care and utilization committees at Memorial Sloan Kettering Cancer Center, New York, NY.

As indicated above, all animal studies were approved by the local authorities and carried out in accordance with the NIH guidelines and the Protection of Animals Act.

**CRISPR/Cas9-mediated gene editing**. CTSB knockout U2OS cell lines were generated by CRISPR/Cas9-mediated genome editing, using predesigned ready-to-use Cas9 and guide RNA sequences that target CTSB exon 2 in U6gRNA-Cas9–2A-GFP (Sigma-Aldrich, ID HS0000248520/1). Selection of transfected cells was based on GFP expression, using flow cytometry (FACSVerse™, BD Biosciences). CTSB deletion in the selected clones was verified by immunoblotting and sequencing.

**Transfections**. *CSTB* coding region was amplified from CSTB cDNA clone IRATp970G1013D (Source Bioscience) with *CSTB* primers (5′-cgccagggatc-caccggtcgccaccatgatgtgcgggcgccctccgccacgcag-3′ and 5′-aaggaaaaaagcggccgctca-gaaataggtcagctcatcatgcttggc-3′ (TAG Copenhagen A/S). PCR products, pBOS-H2B-EGFP (ref.[73]), pBOS-H3-EGFP (Ref.[74]), and pcDNA3.1(+) (Life Technologies) were digested with BamHI-HF® (R3136S) and NotI-HF® (R3189S) restriction enzymes from New England Biolabs, and ligated to generate pBOS-H2B-CSTB, pBOS-H3-CSTB and pcDNA3.1-CTSB expression plasmids. Site directed mutagenesis of pBOS-H3-EGFP construct was performed using the QuikChange Lightning Site-Directed Mutagenesis kit (Thermo Fisher Scientific, C10425) according to manufacturer's instructions using primers 5′-aaa-gagccctaccaaggcggcctcacaagcctcc-3′ and 5′-ggaggcttgtgaggccgccttggtagggctcttt-3′ for Y100A mutation, 5′-caaagagccctaccaactcggcctcacaagcctcc-3′ and 5′-ggaggcttgt-gaggccgagttggtagggctctttg-3′ for Y100E mutation, 5′-agagccctaccaagaaggcctca-caagcc-3′ and 5′-ggcttgtgaggcctttcttcttggtagggctct-3′ for Y100F mutation and 5′-cagccttggtagcccatgcttgcgtggcg-3′ and 5′-cgccacgcaagcagtgggctaccaaggctg-3′ for L21W mutation (TAG Copenhagen A/S). Mutations were verified by sequencing at LGC Genomics GmbH. Transfections were performed in Opti-MEM® I medium (Thermo Fisher Scientific, 31985070) using Lipofectamine LTX Plus reagent for cDNA (Life Technologies, 15338-100) or in complete medium using RNAiMAX Lipofectamine reagent (Life Technologies, 13778150) and 20 nM siRNA or 30–50 pM esiRNA (All Star non-targeting control siRNA from Qiagen (SI03650318), *CTSB* siRNA#1 (5′-gcugaagcuuggaacuucuggacaa-3′) from Invitrogen and *CTSB* siRNA#2 (SASI_Hs01_00108036), *CTSB* siRNA#3 (SASI_Hs01_00108037), *EGFP* MISSION® esiRNA (EHUEGFP) and *TERF2* MISSION® esiRNA (EHU042991) from Sigma-Aldrich) according to the manufacturer's instructions.

**Cell synchronization**. When indicated, cells were synchronized in late G2 phase of the cell cycle by incubation with 9 µM RO3306 for 16 h, washed twice for 5 min in pre-warmed (37 °C) culture medium and allowed to progress for 1–3 h into the desired phase of mitosis in pre-warmed culture medium containing indicated inhibitors at 37 °C.

**Antibodies**. Primary and secondary antibodies used are listed in Supplementary Table 1.

**Immunoblotting**. Cell lysates and chromatin-bound protein extracts were separated in 4–15% or 12% Mini-PROTEAN TGX Gels (BIO-RAD, 456-1086 and 456-1043), transferred to nitrocellulose membrane (BIO-RAD, 170-4158) using Bio-

Rad Trans-Blot Turbo system (Bio-Rad), and stained with indicated primary antibodies and matching secondary HRP-conjugated antibodies listed above. Immunoreactivity was detected following incubation with Clarity Western ECL reagents (Bio-Rad, 170-5061) with Luminescent Image Reader (Fujifilm, LAS-4000), and quantified by densitometry with Image Studio Lite software (LI-COR Biosciences).

**Immunocytochemistry**. Cells were grown on glass coverslips, fixed in 4% paraformaldehyde (PFA, Ampliqon, 43226.1000), permeabilized and blocked in Dulbecco's phosphate-buffered saline (DPBS, Thermo Fisher Scientific, 14190-094) containing 5% goat serum (DAKO, X090), 1% bovine serum albumin (BSA, Amresco, E531) and 0.3% Triton-X-100 (Sigma-Aldrich, T9284), and incubated with the indicated primary and matching Alexa Fluor®-conjugated secondary antibodies listed above. Nuclei were stained with 2.5 µg per mL Hoechst 33342 (Sigma-Aldrich, B2261) and coverslips were mounted with Prolong Gold Antifade mounting medium (Life Technologies, P36930) prior to confocal microscopy or super-resolution structured illumination microscopy (SR-SIM).

**Proximity ligation assay (PLA)**. PLA with intact cells were performed after fixing in 4% PFA and permeabilization in Tris-buffered saline (TBS) containing 5% goat serum, 1% BSA and 0.3% Triton-X-100 using antibodies against CTSB and P-Ser-10-histone H3, DuoLink in situ PLA Probes and Detection Reagents (Sigma-Aldrich, DUO92008, DUO92005, and DUO92001) according to the manufacturer's instructions. Nuclei were stained with 2.5 µg per mL Hoechst 33342 and actin fibers with Alexa Fluor® 488-conjugated phalloidin (Thermo Fisher Scientific, A12379) prior to confocal microscopy. To obtain chromosome spreads, cells were synchronized as described above and released in growth medium containing the indicated drugs and 0.2 µg per mL karyoMAX®/colcemid™ (Thermo Fischer Scientific, 15212-012) for 1 h. Trypsinized cells were then washed in DPBS, resuspended in 75 mM KCl and incubated at 20 °C for 15 min before dropwise addition of 250 µL MeOH/Acetic acid (3:1) on vortex and centrifugation at $1200 \times g$ for 5 min. After removal of the supernatant, 1 mL MeOH/Acetic acid (3:1) was added to the pellet and the sample was incubated for 1 h at 20 °C before centrifugation at $1500 \times g$ for 5 min. The obtained pellet was resuspended in 250 µL MeOH/Acetic acid (3:1) and dropwise placed on Superfrost plus™ slides (Thermo Fischer Scientific, J1800AMNT), air-dried and incubated at 4 °C for a minimum of 14 h before the PLA assay.

**Immunohistochemistry**. Paraffin blocks were processed for immunohistochemistry using standard techniques. Intestine sections (4 µm) were deparaffinized in xylene, boiled in 10 mM trisodium citrate buffer (pH 6.0), and stained with indicated primary antibodies at 22 °C for 2 h, matching Alexa Fluor®-conjugated secondary antibodies at 22 °C for 1 h, and stained with 2.5 µg per mL Hoechst 33342 prior to confocal microscopy. Tumor xenograft and kidney sections (4 µm) were further deparaffinized and rehydrated with descending alcohol series, boiled in Tris-EDTA antigen retrieval buffer (pH 9.0) for 10 min, and blocked in 1% FCS in TBS. Indicated primary antibodies in TBS containing 0,1% Triton-X and 1% BSA were incubated for 16 h at 4 °C and appropriate Alexa Fluor®-conjugated secondary antibodies at 22 °C for 15–60 min with 1 µg per mL Hoechst prior to confocal microscopy.

**Immuno-fluorescence in situ hybridization (ImmunoFISH)**. Cells fixed, permeabilized and immunolabelled for indicated proteins were crosslinked with 4% PFA in DPBS for 20 min and incubated with 1 mg per mL RNase (VWR Life Science, 866) in DPBS for 20 min at 37 °C. Cells were then washed twice in DPBS, dehydrated in 70%, 85%, and 100% EtOH and air-dried before the DNA was denatured at 80 °C in hybridization buffer (10 mM NaHPO$_4$ (pH 7.4), 10 mM NaCl, 20 mM Tris (pH 7.5), 70% formamide and 1% maleic acid buffer (100 mM Maleic acid, 150 mM NaCl, 175 mM NaOH, pH 7.2) with 100 mg per mL blocking reagent (Sigma-Aldrich, 11096176001) containing 800 ng per mL TelC-Alexa488 PNA telomere probe (Panagene, F1004) for 4 min, and cells were incubated for 2 h at 22 °C in a humid and dark chamber. After 10 min wash in DPBS with 0.1% Tween-20 (65 °C) and 1 min wash in 2x SCC buffer (0.3 M NaCl, 30 mM sodium citrate, pH 7.2) with 20% Tween-20, the cells were dehydrated in 70%, 85%, and 100% EtOH and air-dried before the DNA was stained with 2.5 µg per mL Hoechst 33342. The coverslips were mounted on glass slides using Prolong Gold Antifade mounting medium before confocal microscopy.

**Cholesterol staining**. Cells were fixed with 4% methanol-free PFA and incubated with 50 µg per mL filipin III (Sigma-Aldrich, F4767) in DPBS at 37 °C for 30 min. When indicated, cells were additionally stained for LAMP2 and analyzed as described in "immunostaining" section above.

**Confocal microscopy**. Images were captured with a Zeiss LSM700 microscope with Plan-Apochromat 63×/1.40 Oil DIC M27 objective and Zen 2010 software (Carl Zeiss).

**SR-SIM**. Thin (0.1 μm) z-stacks of high-resolution image frames were collected in 5 rotations by utilizing an alpha Plan-Apochromat 100×/1.46 oil DIC M27 ELYRA objective and ELYRA PS.1 Microscope (Carl Zeiss) equipped with a 488 nm laser (100 mW), 561 nm laser (100 mW) and iXon DU 885 EM-CCD camera (Andor Technology). Images were reconstructed using ZEN software (black edition, 2012) based on a structured illumination algorithm[75]. Analysis was performed on reconstructed super-resolution images in ZEN. Maximum-intensity projections were produced and 3-D rendering was performed of selected regions of interest, using a render series of 150 frames.

**Live-cell imaging**. To visualize anaphase bridges, U2OS-H2B-GFP/mCherry-α-tubulin cells were plated on 8-well chambered cover glass (Thermo Scientific, 155409) and imaged in subconfluent conditions. Cells were incubated in media supplemented with 25 mM HEPES and treated as indicated immediately prior to imaging. Cells were imaged throughout cell division with 2-min frame interval. To follow lysosomal permeabilization during mitosis, U2OS-LGALS3-mCherry cells were plated on 8-well chambered coverglass and loaded overnight with 0.5 mg per mL 10 kDa dextran coupled to Alexa Fluor® 488 (Thermo Fisher Scientific, D22910). Excess dextran was washed away and cells were allowed to settle to new medium supplemented with 25 mM HEPES for a minimum of 2 h before imaging. Cells in mitosis were imaged at 1 frame/minute and LGALS3-positive lysosomes in dividing cells were followed until anaphase. All live-cell imaging was performed in a heated chamber (37 °C) using a 60× NA 1.4.a objective mounted on a Perki-nElmer spinning-disc confocal microscope.

U2OS-H2B-GFP cells cultured in 35 mm glass-bottomed dishes (14 mm, No. 1.5, MatTek Corporation) were transfected with 5 nM of siRNA for 72 h for the single transfections, and two times during 96 h for the double transfections prior to time-lapse imaging, which was performed in a heated chamber (37 °C) using a Plan-Apochromat 63×/1.4NA (Carl Zeiss) with differential interference contrast oil objective mounted on an inverted Zeiss Axio Observer Z1 microscope (Marianas Imaging Workstation from Intelligent Imaging and Innovations Inc. (3i), Denver, CO, USA), equipped with a CSU-X1 spinning-disk confocal head (Yokogawa Corporation of America) and three laser lines (488 nm, 561 nm and 640 nm). Images were detected using an iXon Ultra 888 EM-CCD camera (Andor Technology). Eleven 1 μm-separated z-planes covering the entire volume of the mitotic spindle were collected every 2 min. All displayed images represent maximum-intensity projections of z-stacks. Image processing was performed in ImageJ (Fiji)[76].

**Aurora B assay**. U2OS cells were grown on glass coverslips and treated as indicated, fixed for 10 min at 37 °C with 4% PFA diluted in PHEM buffer (60 mM PIPES, 25 mM HEPES, 10 mM EGTA, 4 mM MgSO$_4$, pH 7.0) to stain for AURKB-P-T323[77] or in PTEM buffer (20 mM PIPES, 0.2% Triton-X, 10 mM EGTA, 1 mM MgCl$_2$, pH 6.8) to stain for total AURBK. Cells were stained with indicated primary antibodies and matching Alexa Fluor®-coupled secondary antibodies listed above and DNA was counterstained with 0.1 μg per mL DAPI (Sigma-Aldrich, D3571) before the coverslips were mounted with Fluoromount-G™ solution (Sigma-Aldrich, 00-4958-02) on glass slides. Images were acquired in the Zeiss AxioOb-server Z1 wide-field microscope (63x Plan-Apochromatic oil differential inter-ference contrast objective lens, 1.4 NA) equipped with a CoolSNAP HQ2 CCD camera (Photometrics) using the Metamorph software.

For the quantification of fluorescence intensities at kinetochores, a MATLAB-based algorithm was used .pngt from António Pereira, Instituto de Investigação e Inovação em Saúde, Porto, Portugal). Fluorescence was measured by quantification of pixel gray levels on the different z planes within a region of interest (kinetochore). Co-staining with anti-centromere antibody (ACA) was used to define the kinetochore regions, which were then used to quantify the fluorescence intensity of the protein of interest in the same region. Background fluorescence was measured outside the region of interest and subtracted. Fluorescence intensities of the protein of interest were normalized against centromere signals. Over 1000 kinetochores from 20–40 cells from two independent experiments were analyzed for each protein.

**In vitro histone cleavage assay**. Chromatin was extracted using the Subcellular Protein Fractionation Kit for Culture Cells (Thermo Fisher Scientific, 78840) according to the manufactures instructions with the exception that the provided protease inhibitors were substituted with 0.5 mM pefablock (Sigma-Aldrich, 11429868001). Equal amounts of chromatin (30 μg protein) were incubated at 37 °C for 15 min in a pH-modified cathepsin reaction buffer (50 mM sodium acetate, 4 mM EDTA, pH 7.0), containing 0.5 mM pefabloc, 5 mM dithiothreitol (Sigma-Aldrich, D9779), and indicated amounts of rCTSB (Biovison, 7580), rCTSL (Bio-vison, 1135) and ALLN (Calbiochem, 208719). Samples were subsequently boiled for 5 min and analyzed by immunoblotting.

**Cysteine cathepsin activity assays**. To estimate the total cellular cysteine cathepsin activity in cell lysates, cells were lysed using caspase lysis buffer (25 mM HEPES, 5 mM MgCl2, 1 mM EGTA, 0.5 mM pefablock, 0.1 mM dithiothreitol and 0.025% Triton X buffer; pH 6.0) for 15 min on a shaking table at 4 °C. Cells were then centrifuged for 5 min at 3000 rcf at 4 °C, and 50 μL supernatant was

transferred to a black bottom 96-well plate together with 50 μL substrate buffer (Cathepsin reaction buffer at pH = 6, 8 mM dithiothreitol, 0.5 mM pefablock and 20 mM zFR-AFC (Sigma-Aldrich, P4157)). Lastly, light emission (excitation at 400 nm; emission at 489 nm) was measured on a Varioskan Flash spectral scanning multimode reader (Thermo Fisher Scientific) every 55 s for 20 min at 37 °C. Values were normalized by protein content.

To visualize the cysteine cathepsin acitivity in living cells, U2OS cells were incubated with 0.25 mg per mL 10 kDa dextran coupled to Alexa Fluor® 488 for 16 h, washed and chased for 2 h. Magic Red Cathepsin B substrate (AbD Serotec, ICT937) was added at 1:25 (vol:vol) with indicated treatments and cells were incubated for 15 min at 37 °C before live cell imaging in a heated chamber (37 °C) using a ×60 NA 1.4.a objective mounted on a PerkinElmer spinning-disc confocal microscope. Confocal step size, 400 nM. Extralysosomal CTSB activity at mitotic chromatin was defined as the presence of Magic Red signal outside dextran-positive lysosomes proximal to chromosomes.

**Two-dimensional flow cytometry**. Cell cycle distribution was analyzed using the Click-it EdU Alexa Fluor Flow Cytometry Kit (Thermo Fisher Scientific, C10425) according to the manufacturer's instructions. Briefly, S-phase DNA was stained with 1 μM EdU for 1 h and total DNA with FxCycle™ Far Red, both from the kit. Cells were subjected to flow cytometry (FACSVerse™, BD Biosciences) and ana-lyzed with FlowJo version 10 software (FlowJo, LLC) for cell cycle distribution. The percentages of cells in G1, S and G2/M phases were calculated with FlowJo by gating the populations in a typical horseshoe-shaped dot plot.

**Quantification and statistical analyses**. All experiments were performed a minimum of three times and results are expressed as mean + SD from ≥3 inde-pendent experiments. For all image-based analyses ≥10 randomly chosen cells/sample were analyzed. Multiple comparisons were carried out using one-way ANOVA combined with Dunnett's multiple comparisons test and comparisons of two groups with unpaired, two-tailed Student's t-test. Statistical analyses and graphs were performed using Prism version 7.0a software (GraphPad).

**Reporting summary**. Further information on research design is available in the Nature Research Reporting Summary linked to this article.

## Data availability

All relevant data generated or analyzed during this study are included in this published article (and its Supplementary Information files). The source data underlying Figs. 1a, 1d, 1f, 1g, 2b, 2d, 3a, 3c, 3e, 3f, 3h, 3i, 4a–c, 4e, 4g, 4h, 5c, 5d, 5f–i, 6b, 6d, 6e, 7b–h, and Supplementary Figs 1a, 1g, 1h, 3c-g, 4a, 4d, 4e, 5b, and 7a–c are provided as a Source Data file.

## Material and code availability

The non-commercial materials will be available from the authors after reasonable request. No custom-made codes or mathematical algorithms were used.

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

## Acknowledgements

We thank D. Larsen, L. Vandervox, N. Klemm, T. Dietrich, and M. Nielsen for technical assistance, and Drs. J. Bartek, H. Kimura, S. Geley, T. Kirkegaard, A. Linkermann, H. Leffler, B. Margulis, A. Pereira, G.M. Wahl, and H. Wodrich as well as Developmental Studies Hybridoma Bank for valuable reagents. The research reported in this publication was supported by European Research Council (AdG 340751 to M.J.), Danish National Research Foundation (DNRF125 to M.J.), Danish Cancer Society (R90-A5783 to M.J.; R146-A9322 to M.B.), Novo Nordisk Foundation (NNF15OC0016914 to M.J.), Danish Council for Independent Research (7016-00360B to M.J. and 4004-00287 to E.F.), Federation of European Biochemical Societies (Long term fellowship to E.F.), Lundbeck Foundation (R231-2016-3051 and R233-2016-3797 to E.F.; R215-2015-4081 to M.B.), Orion Research Foundation (to S.H.), Cancer Society of South-West Finland (to S.H.), Instrumentarium Science Foundation (to S.H.), DFG (RE1584/6–2 to T.R.), SFB (850/B7 to T.R.), the Cancer Association South Africa (CANSA to B.Lo) and South African Medical Research Council (SAMRC to B.Lo).

## Author contributions

S.H. made the initial observation leading to the hypothesis. S.H., J.L.S. and E.F. designed and performed the majority of experiments and analyzed the data with great contributions from Q.Y. and L.C. AURKB activity and live cell imaging of *CTSB* depleted cells were analyzed by C.G. and M.B. B.Lo. performed super-resolution microscopy. B.Li assisted with the PLA experiments and study design. T.R. and J.A.J. provided cathepsin deficient murine tissues. M.J. designed the overall study, supervised the experiments, analyzed the data and wrote the first draft of the paper. All authors contributed to the final text and approved it.

## Competing interests

The authors declare no competing interests.

## Additional information

**Peer Review Information** *Nature Communications* thanks Masashi Narita and the other, anonymous, reviewer(s) for their contribution to the peer review of this work. Peer reviewer reports are available.

