## [Peer Review File · Nature Communications]

REVIEWERS' COMMENTS:

Reviewer #2 (Remarks to the Author):

This revised manuscript by Hamalisto et al has addressed several of the concerns expressed in the first round. Nonetheless, several open ends still remain, two of which are particularly important.

The major claims in the paper are highly original, namely that leaky lysosomes drive proper chromosome segregation. The authors show that leaky lysosomes can be detected in a large fraction of mitotic cells, and that these leaky lysosomes associate with chromosomes, primarily in the vicinity of telomeres. The authors also show that suppression of leakiness of the lysosomes results in problems in chromosome segregation, and inversely they have added data to the rebuttal (not yet in the revised manuscript) that perturbation of telomere function will produce more leaky lysosomes. This represents an intriguing set of data, that I feel merits publication.

However, as stated, two important open ends remain.

Firstly, it is unclear why only a subset of mitotic cells display leaky lysosomes. The authors argue that this is because leakiness is somehow triggered by problems in chromosome (telomere) disjunction. Based on the data presented in the manuscript, this is speculation. The data showing that interference with telomere function does produce more leaky lysosomes does lend some support for this speculation and should therefore be included in the paper to strengthen this claim. I would prefer to see that also in this case, cholesterol has an effect on chromosome segregation.

These data will not fully close this gap, but at least it will help the reader to better interpret what might be underlying the observations presented. The authors should remove the statement that the leaky lysosomes can resolve telomere end-to-end fusions, because they do not have any data for this. I do agree with the authors that a deep mechanistic insight in the trigger that produces leaky lysosomes will require a substantial amount of extra work. I don't object to a descriptive paper on an exciting novel phenomenon, but the authors should provide some level of support for the speculative model that is presented.

Secondly, it is not entirely clear what the leaky lysosomes do to promote proper chromosome segregation. The authors show data on histone H3 cleavage, but how cleavage of histone H3 promotes sister chromosome resolution is not clear. The authors favor a model in which the leaky lysosomes promote resolution of entangled or fused telomeres. I suggested earlier that they should analyze metaphase spreads to demonstrate that cells that fail to induce leaky lysosome (cholesterol) contain more fused or entangled telomeres.

The authors argued that this was not feasible, because metaphase spreads in U2OS cells are "very messy" and the phenotype would only present itself in late metaphase or anaphase. I'm not satisfied with this answer and feel that some more evidence on telomere entangling/fusion as the root of the

observed phenotype needs to be presented. The phenotype is also seen in MCF7 cells and several other settings, which could be used to do this? Also, the leaky lysosomes are most abundant in prometaphase, suggesting that they start to act much earlier, and could promote telomere untangling well before anaphase. In fact, if they wouldn't do this, anaphase bridges would be equally prominent in both cases, but would not be resolved in the cells treated with cholesterol.

It would also be very useful for the reader to know how long the galectin puncta persist after leakiness. This information is highly relevant to interpret the appearance/disappearance of puncta-positive lysosomes in the different phases of mitosis, in relation to the proposed mode-of-action. As far as I understand, the puncta persist well after the lysosome has regained integrity, but I assume the leaked out cathepsin is not active for very long periods of time?

Reviewer #3 (Remarks to the Author):

The authors have addressed most of the reviewers' questions. They have added a substantial amount of new data/experiments. I have no additional questions.

Reviewer #4 (Remarks to the Author):

The authors have done a good job addressing the reviewer's comments. This manuscript describes a very intriguing new role of lysosomes in maintenance of genome integrity and chromosome segregation. Overall, the central conclusions of the study are solidly grounded in abundant and well-executed experiments. Although there is a number of important questions that remain unanswered (e.g. what determines which lysosomes will release cathepsins during metaphase, what is the mechanism of leakage induction, how different lysosomal enzymes contribute to telomere segregation), the study contains, in my opinion, sufficient novel information to grant publication. I just have a couple of minor comments:

1. In order to provide additional mechanistic information that further support the conclusions of the study, I would recommend including in the manuscript the data showing that induction of increased telomere stress by TERF2 depletion increases the number of chromatin-associated leaking lysosomes (rebuttal letter-Figure 1).

2. Several of the panels in Supplemental Figure 1 are not properly labeled. For example, negative BAX staining in cells containing LGALS3-positive lysosomes is shown in Fig. S1g (not Fig. S1c as mentioned in the main text-page 6); absence of LC3 staining is shown in Fig. S1h (not Fig. S1d as

stated in page 6), and quantification of leaky lysosomes in HEK293 cells is shown in Fig. S1f (not Fig. S1h-page 7).

REVIEWERS' COMMENTS:

Reviewer #2 (Remarks to the Author):

This revised manuscript by Hamalisto et al has addressed several of the concerns expressed in the first round. Nonetheless, several open ends still remain, two of which are particularly important.

The major claims in the paper are highly original, namely that leaky lysosomes drive proper chromosome segregation. The authors show that leaky lysosomes can be detected in a large fraction of mitotic cells, and that these leaky lysosomes associate with chromosomes, primarily in the vicinity of telomeres. The authors also show that suppression of leakiness of the lysosomes results in problems in chromosome segregation, and inversely they have added data to the rebuttal (not yet in the revised manuscript) that perturbation of telomere function will produce more leaky lysosomes. This represents an intriguing set of data, that I feel merits publication.

However, as stated, two important open ends remain.

Firstly, it is unclear why only a subset of mitotic cells display leaky lysosomes. The authors argue that this is because leakiness is somehow triggered by problems in chromosome (telomere) disjunction. Based on the data presented in the manuscript, this is speculation. The data showing that interference with telomere function does produce more leaky lysosomes does lend some support for this speculation and should therefore be included in the paper to strengthen this claim. I would prefer to see that also in this case, cholesterol has an effect on chromosome segregation.

These data will not fully close this gap, but at least it will help the reader to better interpret what might be underlying the observations presented. The authors should remove the statement that the leaky lysosomes can resolve telomere end-to-end fusions, because they do not have any data for this. I do agree with the authors that a deep mechanistic insight in the trigger that produces leaky lysosomes will require a substantial amount of extra work. I don't object to a descriptive paper on an exciting novel phenomenon, but the authors should provide some level of support for the speculative model that is presented.

As requested by the reviewer, we have included data showing that the interference with telomere function by depletion of TERF2 induces a significant increase in the number of pro(metaphases) with chromatin-proximal leaky lysosomes as well as in the number of chromatin-proximal leaky lysosomes per cell (Revised Figure 5e-i). In addition, we have tried to test the effect of cholesterol in this setting, but unfortunately the combination of cholesterol loading and siRNA transfection was highly toxic to the cells. Finally, we have modified the statement regarding the chromatin-associated CTSB resolving telomere end-to-end fusions.

Secondly, it is not entirely clear what the leaky lysosomes do to promote proper chromosome segregation. The authors show data on histone H3 cleavage, but how cleavage of histone H3 promotes sister chromosome resolution is not clear. The authors favor a model in which the leaky lysosomes promote resolution of entangled or fused telomeres. I suggested earlier that they should analyze metaphase spreads to demonstrate that cells that fail to induce leaky lysosome (cholesterol) contain more fused or entangled telomeres.

The authors argued that this was not feasible, because metaphase spreads in U2OS cells are "very messy" and the phenotype would only present itself in late metaphase or anaphase. I'm not satisfied with this answer and feel that some more evidence on telomere entangling/fusion as the root of the observed phenotype needs to be presented. The phenotype is also seen in MCF7 cells and several other settings, which could be used to do this? Also, the leaky lysosomes are most abundant in prometaphase, suggesting that they start to act much earlier, and could promote telomere untangling well before anaphase. In fact, if they wouldn't do this, anaphase bridges would be equally prominent in both cases, but would not be resolved in the cells treated with cholesterol.

We performed the suggested experiment using Dual-FISH to detect telomeres and centromeres in metaphase chromosome spreads, but could not see significant changes in numbers of telomere puncta, sister chromatid fusions, dicentric chromosomes, chromatid-type fusions or signal-free ends. In the meanwhile, we realized that disturbance of microtubules by colchicine (used to make metaphase chromosome spreads) or nocodazole inhibited lysosomal leakage in mitosis. Thus, inhibition of lysosomal leakage by cholesterol is not likely to increase telomere defects in metaphase spreads prepared using standard methods.

It would also be very useful for the reader to know how long the galectin puncta persist after leakiness. This information is highly relevant to interpret the appearance/disappearance of puncta-positive lysosomes in the different phases of mitosis, in relation to the proposed mode-of-action. As far as I understand, the puncta persist well after the lysosome has regained integrity, but I assume the leaked out cathepsin is not active for very long periods of time?

We have revised the introduction to indicate that galectin puncta can persist several hours as shown in Ref. 39.

Reviewer #3 (Remarks to the Author):

The authors have addressed most of the reviewers' questions. They have added a substantial amount of new data/experiments. I have no additional questions.

Reviewer #4 (Remarks to the Author):

The authors have done a good job addressing the reviewer's comments. This manuscript describes a very intriguing new role of lysosomes in maintenance of genome integrity and chromosome segregation. Overall, the central conclusions of the study are solidly grounded in abundant and well-executed experiments. Although there is a number of important questions that remain unanswered (e.g. what determines which lysosomes will release cathepsins during metaphase, what is the mechanism of leakage induction, how different lysosomal enzymes contribute to telomere segregation), the study contains, in my opinion, sufficient novel information to grant publication. I just have a couple of minor comments:

1. In order to provide additional mechanistic information that further support the conclusions of the study, I would recommend including in the manuscript the data showing that induction of increased telomere stress by TERF2 depletion increases the number of chromatin-associated leaking lysosomes (rebuttal letter-Figure 1).

As requested by the reviewer, we have included data showing that the interference with telomere function by depletion of TERF2 induces a significant increase in the number of pro(metaphases) with chromatin-proximal leaky lysosomes as well as in the number of chromatin-proximal leaky lysosomes per cell (Revised Figure 5e-i).

2. Several of the panels in Supplemental Figure 1 are not properly labeled. For example, negative BAX staining in cells containing LGALS3-positive lysosomes is shown in Fig. S1g (not Fig. S1c as mentioned in the main text-page 6); absence of LC3 staining is shown in Fig. S1h (not Fig. S1d as stated in page 6), and quantification of leaky lysosomes in HEK293 cells is shown in Fig. S1f (not Fig. S1h-page 7).

We apologize the mistakes in labelling – the errors have been corrected in the revised manuscript.